# Key homeobox transcription factors regulate the development of the firefly's adult light organ and bioluminescence

**Xinhua Fu** [1] ✉ **& Xinlei Zhu** [2]

Adult fireflies exhibit unique flashing courtship signals, emitted by specialized light organs, which develop mostly independently from larval light organs during the pupal stage. The mechanisms of adult light organ development have not been thoroughly studied until now. Here we show that key homeobox transcription factors AlABD-B and AlUNC-4 regulate the development of adult light organs and bioluminescence in the firefly *Aquatica leii*. Interference with the expression of *AlAbd-B* and *AlUnc-4* genes results in undeveloped or non-luminescent adult light organs. AlABD-B regulates *AlUnc-4*, and they interact with each other. AlABD-B and AlUNC-4 activate the expression of the luciferase gene *AlLuc1* and some peroxins. Four peroxins are involved in the import of AlLUC1 into peroxisomes. Our study provides key insights into the development of adult light organs and flash signal control in fireflies.

The remarkable flashing courtship display of adult fireflies (Lampyridae) has captured the attention of both experts and laymen, turning such exhibitions into popular tourist attractions worldwide[1,2]. The biochemical understanding of firefly luminescence, which involves ATP, $Mg^{2+}$, and $O_2$-dependent luciferase-mediated oxidation of the substrate luciferin[3], along with the cloning of the luciferase gene[4], has led to the widespread use of luciferase as a reporter gene with diverse applications in biomedical research and industry[5].

Glowing or flashing light signals of fireflies are emitted by unique light organs. In larvae, paired light organs with aposematic purpose (repelling predators) form on the eighth abdominal segment[6]. In adult fireflies, light organs are located on the ventral surface of ventrites 6 and 7 in males, and ventrite 6 in females. A distinguishing feature of fireflies in the subfamily Luciolinae is the presence of eight visible abdominal tergites, with the abdomen terminating at ventrite 7, the second of the two light organ-bearing segments in males[7]. Adult light organs consist of three distinctive layers: cuticle, photogenic layer, and dorsal layer or uric acid layer[3]. Multiple photocytes or luminescent cells surround the tracheal end cells (TEC). The photocytes are abundant in mitochondria and peroxisomes containing luciferases[8]. Adult light organs develop independently from larval light organs during the pupal stage, so the paired larval light organs typically persist and function during the pupal stage and the first 24 h after the emergence of adults (Fig. 1A). Two previous reports demonstrated that transcript depletion of the Abdominal-B gene resulted in extensive disruption of the adult light organ development in fireflies[9,10], but there were no further studies on the development of adult light organs. We hypothesized that the key factors influencing the development of adult light organs and bioluminescence in fireflies are the expression of the luciferase gene, and the transportation of luciferase to the reaction organelle of luminescent cells.

In this study, we present a chromosome-level genome assembly for *Aquatica leii* (Fu et Ballantyne), a rare aquatic firefly, using single-molecule Nanopore sequencing and high-throughput chromosome conformation capture sequencing technologies. We also present the functional studies and analyses of two key homeobox transcription factors regulating the adult light organ development in *A. leii*.

## Results

### Chromosome-level genome sequencing and assembly

We sequenced the genome of a male *A. leii* specimen using a combination of single-molecule real-time sequencing (using an Oxford Nanopore Promethion platform), paired-end sequencing (with an MGI2000 short-read sequencing platform), and Hi-C sequencing (Supplementary Table 1). This multifaceted approach allowed us to produce a high-quality chromosome-level genome assembly. Via the

[1]College of Plant Science and Technology, Huazhong Agricultural University, Wuhan 430070, China. [2]Firefly Conservation Research Centre, Wuhan 430070, China. ✉e-mail: fireflyfxh@mail.hzau.edu.cn

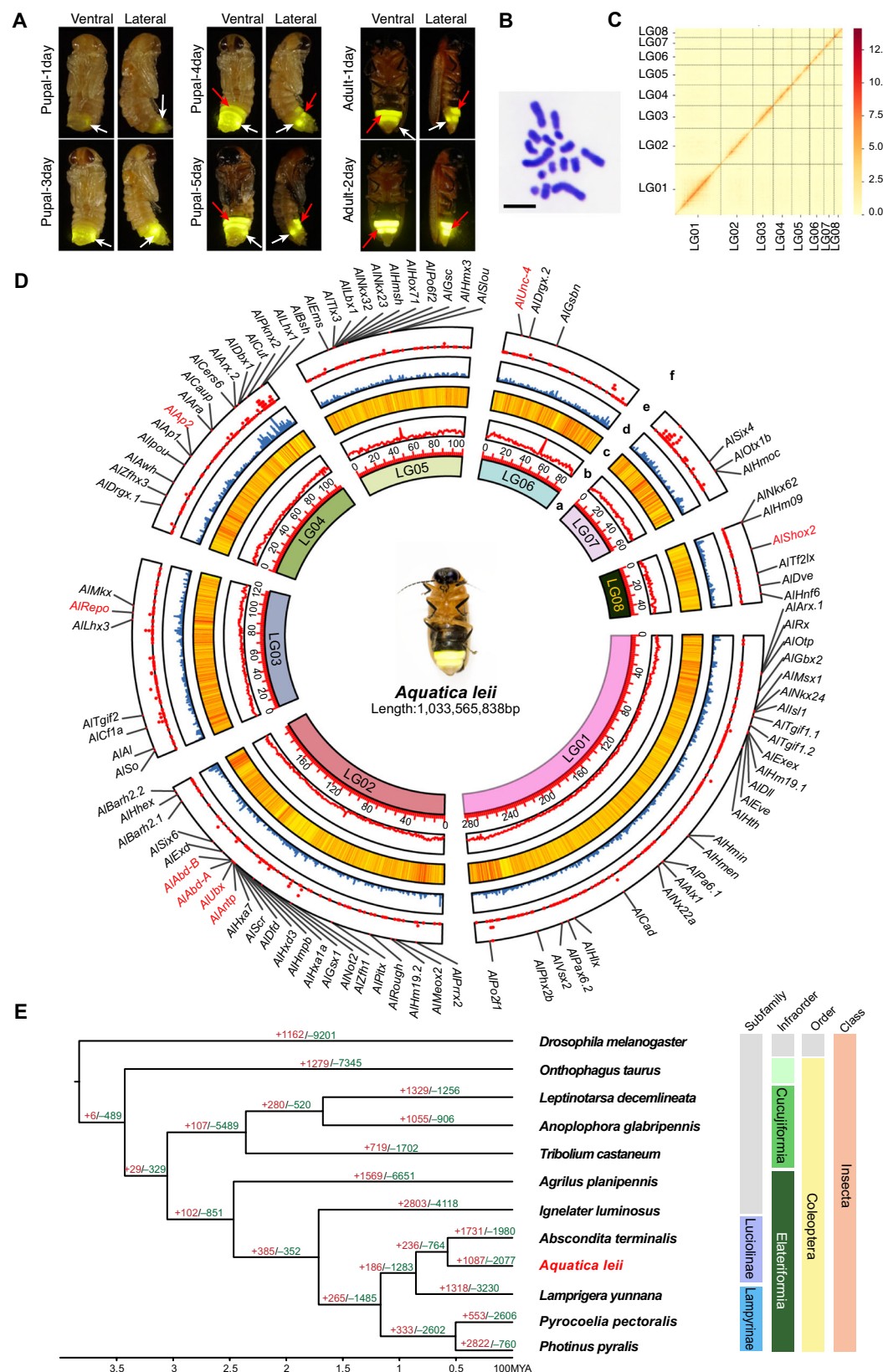

K-mer analysis, we estimated the size of the *A. leii* genome to be 1.09 Gb, with a high level of heterozygosity (3.20%) (Supplementary Fig. 1). The assembled genome was 1.04 Gb in size (Supplementary Table 2), or 95.4% of the estimated genome size. It contained 222 contigs, with the scaffold and contig N50 values of 125.64 Mb and 10.81 Mb, respectively (Supplementary Table 2). We also analyzed the

chromosome number and karyotype ($2n = 14 + XY$) (Fig. 1B). Hi-C scaffolding of the genome resulted in 191 scaffolds anchored to 8 pseudochromosomes (Fig. 1C, D, Supplementary Table 2). We assessed the completeness and quality of the assembly using both genome and transcriptome datasets. The evaluation of genome quality using BUSCO version 5.1.3 and the insecta_odb10 database indicated that

**Fig. 1 | The assembly and evolution of *Aquatica leii* genome. A** Developmental stages of *A. leii*. From top to bottom: pupae aged 1 day to adults aged 2 days (ventral and lateral views). The white arrow heads indicate the larval light organs. The red arrow heads indicate the adult light organs. **B** Karyotype analysis of *A. leii* genome ($2n = 14 + XY$; scale bar = 5 μm), $n = 7$ cells. **C** The genome-wide Hi-C interaction maps of 8 pseudo-chromosomal linkage groups in *A. leii*. **D** Circos plot of genome features. The statistic was based on a 100 Kb window. (a) The inner-most ring shows 8 chromosomes; (b) GC content; (c) Repeat density; (d) Gene density; (e) Transcription factor density; (f) Homeobox family genes. The genes that were functionally verified in subsequent experiments are marked in red. **E** Phylogenetic tree of *A. leii* and eleven other insect species. Numbers on the tree indicate the number of gene orthogroup expansions (+) or contractions (−). The divergence time estimates are displayed below the phylogenetic tree. Source data are provided as a Source Data file.

98.2% out of the 1342 genes predicted in the *A. leii* genome by BUSCO were successfully identified and complete. Most of these genes were single-copy loci with 0.3% fragmented and 1.5% missing BUSCOs (Supplementary Table 3).

A total of 1,985,095 transposable elements (TEs) were detected, constituting approximately 52.93% of the sequence (Fig. 1D). A combined structure- and homology-based analysis identified a total of 603.68 Mb of repetitive sequences, accounting for 58.22% of the complete genome (Supplementary Data 1). This is the highest level among the published firefly genomes (Supplementary Table 4), but we should stress here that this outcome underscores the high quality of the *A. leii* genome assembly, as less precise methods often fail to assemble repetitive DNA into scaffolds[11].

By integrating ab initio prediction evidence, homology prediction evidence, and RNA-sequencing (RNA-seq) evidence, we identified 16,472 protein-coding genes with high confidence and precision. Among them, the average gene size was 32,544 bp, CDS length was 1478 bp, and intron length was 6719 bp (Supplementary Table 5). The *A. leii* genome is the largest among the currently known firefly genomes. Similarly, the number of introns is also larger than in other known firefly genomes, as well as other reference insect genomes used in this study (Supplementary Table 5). This is expected, as these two parameters are correlated[12]. In summary, 14,874 genes (90.30%) were functionally annotated using at least one public database (Swissprot, NR, KEGG, GO, and KOG) (Supplementary Table 6). In addition, four types of ncRNAs were identified, including 156 rRNAs, 2023 microRNAs (miRNAs), 678 cis-regulatory elements, and 661 transfer RNAs (tRNAs) (Supplementary Table 6).

To infer the evolutionary history and phylogenetic relationships of *A. leii*, gene family clustering was performed using OrthoMCL. For this analysis, we also used four other published firefly genomes, six selected coleopteran genomes, and the fruit fly genome as an outgroup (Supplementary Table 7). A total of 17,166 orthologs were identified in these twelve species, 4727 of which were shared by all species (Supplementary Fig. 2). The genome of *A. leii* contained 170 unique orthologs, not shared by any other species. In addition, 1633 single-copy orthologs were identified across these twelve species. These were aligned to develop a super-sequence for each species, subsequently used to construct a phylogenetic tree (Fig. 1E).

Phylogenetic inference indicated that *A. leii* and *Abscondita terminalis* (both Luciolinae subfamily) were sister lineages, as were *Pyrocolia pectoralis* and *Photinus pyralis* (both Lampyrinae subfamily). *Lamprigera yunnana* was the sister taxon to Luciolinae. Divergence time estimation calculated using mcmctree suggested that *A. leii* diverged from the common ancestor of the other members in the subfamily Luciolinae approximately 57.38 million years ago (Fig. 1E).

### Homeobox transcription factors and the development of light organ

A total of 914 transcription factors were identified in the *A. leii* genome. Transcription factor superfamily members were categorized into 45 different classes according to the functional domain characteristics (Supplementary Table 8). After the "zinc finger" transcription factor type, "homeobox" was the second most numerous type of transcription factors. The large group of homeobox genes encode the DNA-binding homeodomain that plays a key role in the development and cellular differentiation during embryogenesis in animals[13]. We undertook an in-depth exploration of homeobox genes on a genomic scale to identify the potential homeobox genes that might govern the developmental processes of adult firefly light organs in *A. leii*. Phylogenetic analysis was conducted using 105 known homeobox genes of *Tribolium castaneum* and 94 genes identified in the *A. leii* genome (Supplementary Fig. 3). All homeobox genes in *A. leii* genome were renamed according to the homology analysis and classified (Supplementary Fig. 3 and Supplementary Data 2).

Statistics and classification of homeobox genes in genomes of 10 model species and *A. leii* firefly conducted using the homeoDB database[14] revealed that the largest number of homeobox genes belonged to the ANTP family (Supplementary Table 9). In the *A. leii* genome, we found 44 homeobox genes belonging to the ANTP family (20 from the HOXL class and 24 from the NKL class). The second largest number of homeobox genes belonged to the PRD family. PROS and HNF families of homeobox genes were not identified in the *A. leii* genome (Supplementary Table 9 and Supplementary Fig. 3). Homeobox genes were identified in each chromosome of *A. leii*: 25 genes in chromosome LG01, 23 in chromosome LG02, 7 in chromosome LG03, 15 in chromosome LG04, 11 in chromosome LG05, 6 in chromosome LG06 (all 6 belonged to the PRO family), 3 in chromosome LG07, and 6 in chromosome LG08 (Fig. 1D and Supplementary Fig. 4 and Supplementary Data 2). As regards the distribution of the homeobox genes families, the ANTP-HOXL family genes were largely concentrated on chromosome LG02, whereas the ANTP-NKL family genes were concentrated on chromosome LG05. In addition, genes of both families were encoded on chromosomes in arranged clusters (Fig. 1D and Supplementary Fig. 4).

The expression patterns of homeobox genes were analyzed using transcriptome data generated from tissue samples of ventrites 6 and 7 collected from male pupae of *A. leii* (these ventrites correspond to the adult light organs in different developmental stages; Supplementary Fig. 5A). Only six homeobox genes were continuously upregulated ($p < 0.05$) during the pupal development, while other genes were downregulated or their regulation shifted from up-regulation to down-regulation during the pupal development (Supplementary Fig. 5A).

The relative expression levels of *AlAbd-A, AlAbd-B, AlUbx, AlAntp, AlUnc-4, AlShox2, AlRepo*, and *AlAp2* were analyzed *in* three developmental stages of female light organs using qPCR (Supplementary Fig. 5B). *AlAbd-B* and *AlUnc-4* were continuously upregulated during the female pupal development, while *AlAbd-A, AlAntp* and *AlUbx* were downregulated. The expression pattern of *AlAp2* had a peak expression in the mid-pupal stage, and the expression levels of *AlRepo* did not change during these three stages. Furthermore, the expression level of *AlShox2* could not be detected (Cq > 37) in female light organs.

We then applied RNAi in both male and female one-day-old pupae to study functions of the six upregulated homeobox genes: *AlAbd-B, AlAntp* (ANTP-HOXL family), *AlUnc-4, AlShox2, AlRepo* (PRD family), and *AlAp2* (LIM family). In addition, we studied functions of two genes, *AlAbd-A* and *AlUbx* (ANTP-HOXL family), tandemly arranged on the same chromosome (LG02) and in the same gene cluster as *AlAbd-B* and *AlAntp* (Fig. 1D). Firefly specimens injected with the double-stranded RNA (dsRNA) of GFP (*dsGfp*) were used as the control group (Fig. 2A).

Phenotypes were also recorded and analyzed in both male and female 2-day-old adults, with a focus on the relative light intensity and

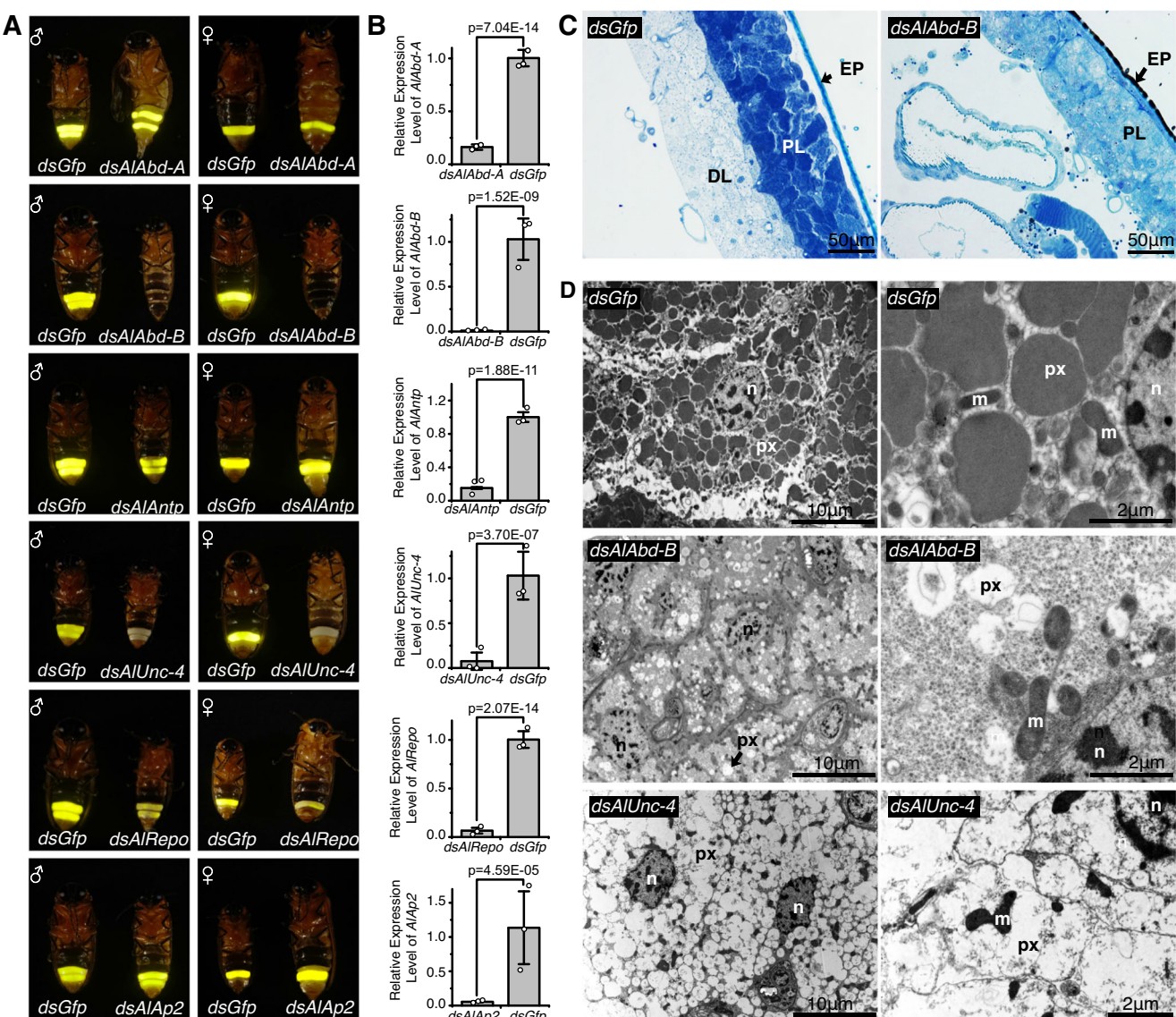

**Fig. 2 | Functional verification of six homeobox genes related to the development of light organs and flash behavior of *A. leii*. A** Functional verification of homeobox genes. Six major genes were knocked down using RNAi, and experimental groups compared to the control group, injected *Gfp* dsRNA (*dsGfp*) (male-left/female-right). **B** Bar graphs show qPCR analysis of *AlAbd-A*, *AlAbd-B*, *AlANTP*, *AlUnc-4*, *AlRepo*, and *AlAp2* in the indicated RNAi samples compared to the *dsGfp* control. The data represent averages ± SD of three biological replicates (2 males and 1 female). The statistical significance was derived using two-sided Student's *t*-tests. **C** Observation of adult light organs by semi-thin sectioning after the knockdown of *AlAbd-B* (scale bar = 50 μm), *n* = 5 independent experiments, EP epidermis, PL photogenic layer, DL dorsal layer. **D** Observation of adult light organs by ultrathin sectioning after the knockdown of *AlAbd-B* and *AlUnc-4* genes (*dsGfp* as the control); scale bars of the left and right image columns are 10 and 2 μm respectively, *n* = 5 independent experiments; n nucleus, px peroxisomes, m mitochondrion. Source data are provided as a Source Data file.

flash patterns, and experimental groups were statistically compared to the control group. The interference efficiency of RNAi was verified by the fluorescence quantitative PCR (qPCR). In 2-day-old adults, we observed a significant decrease ($p < 0.01$) in expression levels of the aforementioned homeobox genes (Fig. 2B). Transcript depletion of *AlAbd-A* (ANTP family - HOXL class) in *A. leii* fireflies resulted in ectopic ventral cuticle depigmentation (Fig. 2A *dsAlAbd-A* group), but it produced no significant differences in light intensity and flash rate (Supplementary Fig. 6A, B and Supplementary Movie 1). This is consistent with previous research[9]. Pupal-stage *AlAbd-B* knockdown (*dsAlAbd-B*) resulted in profoundly disrupted light organ development, ectopically pigmented abdominal cuticle, and the complete loss of luminescence (Fig. 2A, Supplementary Fig. 6A, B and Supplementary Movie 1), which is also consistent with previous research[9,10]. Ultrastructural observations of ventrite 6, which corresponds to the adult light organ,

revealed that peroxisomes decreased in size, and empty peroxisomes were observed in photocytes of the *AlAbd-B* RNAi adult phenotype (Fig. 2C). The whole dorsal layer (urate layer) disappeared completely (Fig. 2D). The depletion of *AlUbx* transcript resulted in pale yellow hindwings, but it did not produce any significant impacts on the development of adult light organs and flash behavior (Supplementary Fig. 6A–D and Supplementary Movie 1). Pupal *AlAntp* RNAi (*dsAlAntp*) treatment resulted in a continuously glowing phenotype, but it did not result in a significant change in light intensity (Fig. 2A, Supplementary Fig. 6A, B and Supplementary Movie 1). Examination of the *dsAlAntp* phenotype in this study suggested that AlANTP regulated the development of neurons involved in flash control.

Transcript depletion of *AlUnc-4* (PRD family) in *A. leii* resulted in a non-luminescent adult phenotype (Fig. 2A *dsAlUnc-4*, Supplementary Fig. 6A, B and Supplementary Movie 1). Ultrastructural observations of

the adult light organs in the *dsAlUnc-4* group revealed shrunken and empty peroxisomes in photocytes (Fig. 2D). Pupal *AlRepo* knockdown resulted in a continuously glowing phenotype and a significant decrease in light intensity (Fig. 2A, Supplementary Fig. 6 and Supplementary Movie 1). *AlShox2* knockdown did not affect light organ formation and flash behavior in *A. leii* (Supplementary Fig. 6A, B, E, F and Supplementary Movie 1).

Pupal *AlAp2* RNAi (LIM family) resulted in a continuously glowing phenotype and a significant decrease in light intensity (Fig. 2A, Supplementary Fig. 6A, B and Supplementary Movie 1).

Overall, based on the fact that the knockdown of homeobox genes *AlAbd-B* and *AlUnc-4* resulted in non-luminescence and empty peroxisomes, these two genes may be the key regulators required for normal light organ development. In addition, we found evidence that *AlAntp*, *AlRepo,* and *AlAp2* are involved in flash control.

## AlABD-B and AlUNC-4 affect the expression of luciferase and pex genes

To further study the regulation of the development of adult firefly light organs, we focused on AlABD-B and AlUNC-4 as putatively key transcription factors. There are two key steps in the development of adult firefly light organs and luminescence. One is the expression of the luciferase gene, and the other is the transportation of luciferase to the reaction organelle, where bioluminescence occurs. Peroxisomal targeting of luciferase was previously documented in mammalian cells[15], yet conclusive evidence supporting the peroxisomal targeting of luciferase in the photocytes of firefly light organs has remained absent until now. We hypothesized that interference with luciferase gene expression or transportation of luciferase resulted in empty peroxisomes. To test this hypothesis, we compared transcriptomes of 1-day-old male adults that underwent the knockdown of *AlAbd-B* and *AlUnc-4* in the pupal stage to the control group (*Gfp* injection). Results showed that the expression of *AlLuc1* decreased significantly in *dsAlAbd-B* and *dsAlUnc-4* knockdown groups (Fig. 3A). The product of *AlLuc1* is a type of a PTS1 peroxisome matrix protein.

The expression of genes related to the transportation of matrix proteins, such as *AlPex5, AlPex13, AlPex14,* and *AlPex1*[16] decreased significantly in the *dsAlAbd-B* group (Fig. 3A). Similarly, the expression of closely related genes, *AlPex3, AlPex16,* and *AlPex19,* which might be related to the transportation of Peroxisome Membrane Protein (PMP)[17,18], also decreased significantly in the *dsAlAbd-B* group. The expression of *AlPxmp2*, possibly related to the nonspecific transportation of small molecules[19], decreased significantly in the *dsAlAbd-B* and *dsAlUnc-4* groups. However, the expression of *AlPx11b* was not changed significantly (Fig. 3A). The expression of *AlPmp34*, responsible for the transport of ATP[20,21], did not change significantly in the *dsAlAbd-B* and *dsAlUnc-4* groups. Different expression patterns of two *Px11c* genes (*AlPx11c.1* and *AlPx11c.2*) related to peroxisomal fission and division[22] were detected in both *dsAlAbd-B* and *dsAlUnc-4* groups: while the expression of *AlPx11c.1* did not change significantly, the expression of *AlPx11c.2* decreased significantly (Fig. 3A, B).

Candidate genes for more detailed analyses were selected using two criteria: high expression level (in the top 10% of genes) and the change in expression $-\log_2 FC$ (*dsAlAbd-B/dsGfp*) > 1.5. The selected genes comprised *Alluc1, AlPx11c.1, AlPx11c.2, AlPex5, AlPxmp2, AlPex13, AlPex14, AlPex16,* and *AlPex1* (Fig. 3B). The RNA-seq analysis of and verification by real-time qPCR revealed that: (1) the expression levels of *AlLuc1, AlPx11c.2, AlPex5* and *AlPxmp2* decreased significantly in response to the knockdown of both homeobox genes (*AlAbd-B* and *AlUnc-4*); (2) the expression levels of *AlPex1, AlPex13, AlPex14,* and *AlPex16* decreased significantly in response to the knockdown of *AlAbd-B* (Fig. 3B); (3) Surprisingly, the expression level of *AlUnc-4* decreased significantly in response to the knockdown of *AlAbd-B* (Fig. 3B). Among these genes, we selected two luciferase genes (*AlLuc1* and its homolog *AlLuc2*) and seven peroxin genes (*AlPx11c.2, AlPxmp2,*

*AlPex5, AlPex13, AlPex14, AlPex16,* and *AlPex1*) for detailed functional studies.

## Functional analyses of luciferase genes *AlLuc1* and *AlLuc2*

Two luciferase genes, *AlLuc1* and *AlLuc2*, were identified in the *A. leii* genome. *AlLuc1* is located within a cluster of fatty acyl-CoA synthetase genes (ACSs) on chromosome LG01 (Fig. 4A). *AlLuc1* has a relatively high homology with ACS genes. The genomic architecture of firefly luciferases and closely related paralogs supports a pervasive role of tandem duplication and neofunctionalization in the birth of this gene family[23]. The structure of *AlLuc1* gene is highly conserved in Lampyridae, with 7 exons in all orthologues (Fig. 4A). In contrast to *AlLuc1*, *AlLuc2* is located on chromosome LG06 and no other ACS genes were located nearby. A phylogenetic tree constructed using AlLUC1 and AlLUC2 proteins, as well as other known proteins from LUC1 and LUC2 families, rendered both families monophyletic (Fig. 4B), which is consistent with previous studies[23,24]. The subcellular localization analysis revealed that AlLUC1 was located in the cytoplasm and nucleus, and AlLUC2 in the peroxisome (Fig. 4C). These results are not consistent with previous research[4,15,25].

Whereas *AlLuc1* was specifically expressed in light organs (Supplementary Fig. 7A), *AlLuc2* had the highest expression in female abdominal tissue (excluding the light organ) (Supplementary Fig. 7B). These findings are consistent with previous research[26]. As opposed to the control group, where the whole body of pupal *A. leii* emitted a weak continuous glow, the *AlLuc2* knockdown resulted in luminescence only in the pupal light organs (Supplementary Fig. 7C). As these results indicated that the development of adult light organs is not associated with *AlLuc2*, we focused on *AlLuc1*.

To test our hypothesis that nonfunctional peroxisomes were directly caused by the lack of AlLUC1 protein, we applied RNAi to the *AlLuc1* gene. This produced a non-luminescent phenotype (Fig. 4D and Supplementary Movie 2). Verification by real-time qPCR confirmed that the expression level of *AlLuc1* was downregulated significantly in non-luminescent adults (Supplementary Fig. 8A). The ultrastructural analysis of the adult light organ in *dsAlLuc1* treatment individuals revealed that approximately half of the peroxisomes had shrunk or were empty (Fig. 4D), which is similar to the effects of *AlAbd-B* RNAi and *AlUnc-4* RNAi. To exclude the possibility that a lack of luciferin could lead to shrunken or empty peroxisomes, five genes responsible for the synthesis and regeneration of luciferin were analyzed by *dsAlAbd-B* and *dsAlUnc-4* RNA-seq: B-glucosidase genes (BGL), phenoloxidases (PO), acyl-CoA thioesterases (ACOT), luciferases (LUC), and luciferin regenerating enzymes (LRE)[27]. Results revealed that expression levels were not significantly changed, except for *AlLuc1*, which decreased significantly (Supplementary Fig. 9). Furthermore, the content of luciferin was not significantly changed ($P > 0.01$) in the *AlAbd-B* RNAi group compared with the control group (Supplementary Fig. 10). We conclude that the lack of luciferase in peroxisomes of the adult light organ caused non-luminescence. Our research offers the evidence that the LUC1 type of luciferase is localized to peroxisomes within the firefly photocytes.

## AlABD-B and AlUNC-4 can transactivate expression of *AlLuc1* in heterologous systems

The expression level of *AlLuc1* was significantly downregulated in transcriptomes of *dsAlAbd-B* and *dsAlUnc-4* groups. Via the JASPAR database analysis, two putative DNA-binding domains were identified for each transcription factor: AlABD-B (−145 and −415, negative numbers indicate that it is upstream of the transcription start site) and AlUNC-4 (−492 and −557) (Fig. 4A). We hypothesized that AlABD-B and AlUNC-4 interacted with the *AlLuc1* promoter and upregulated its activity, and carried out multiple assays to test this hypothesis: yeast one-hybrid (Y1H), electrophoretic mobility shift assays (EMSA), a dual-luciferase reporter assay, Western blot, and Immunofluorescence (IF) assay.

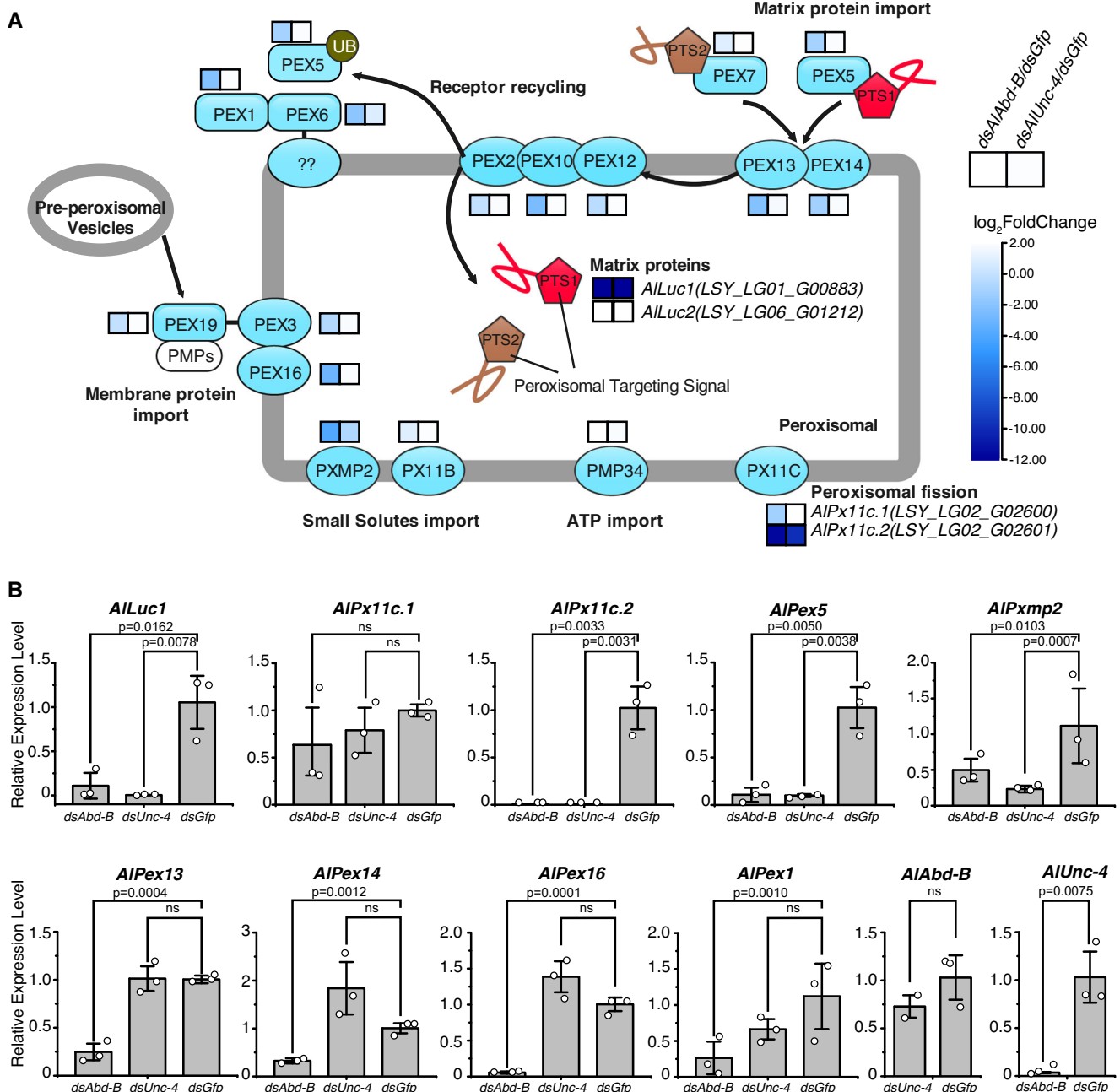

**Fig. 3 | Differential gene expression analysis of transcriptomes of *AlAbd-B* RNAi and *AlUnc-4* RNAi phenotypes compared to the control group. A** The effect of knockdown of *AlAbd-B* and *AlUnc-4* on expression profiles of Pex genes (transcriptome analysis). A representation of peroxisome biogenesis pathways in *A. leii* (modified ko04146 in *Drosophila*)[71] (image published with permission from KEGG Database Project, Kanehisa Laboratories, Japan, and the original version can be found https://www.kegg.jp/pathway/ko04146). Heatmap shows the expression levels of differentially expressed Pex genes in *dsAlAbd-B* and *dsAlUnc-4* groups. The color key corresponds to the log₂FC (fold change of *dsRNA/dsGfp*) heatmap of each gene in two samples (*n* = 3). **B** The expression levels of ten genes in *AlAbd-B* RNAi and *AlUnc-4* RNAi groups compared with the *dsGfp* control inferred using qPCR. All data were calculated as the mean value of three replicates (2 males and 1 female). Error bars represent standard deviations. The data were statistically analyzed using one-way ANOVA and Tukey's multiple range tests, where ns stands for not significant. Source data are provided as a Source Data file.

A Y1H assay technique was performed to confirm the interaction between the transcription factor (AlABD-B or AlUNC-4) and the promoter of the *AlLuc1* gene (*proAlLuc1*) (Fig. 4E, F). The yeast one-hybrid results showed that Y1H Gold yeast cells containing *pABAi-proAlluc1* and pGADT7-AlABD-B grew on the SD/-Leu/-Ura/AbA selective medium. Y1H Gold bait yeast strains containing *pABAi-proAlluc1-mt1* (AlABD-B binding sites deletion mutant) and pGADT7-AlABD-B were not able to grow on a selective medium. None of the negative control Y1H Gold bait yeast strains could grow on the aureobasidin A (AbA)

selective medium (Fig. 4E). Y1H assay between AlUNC-4 and *proAlluc1* or *pABAi-proAlluc1-mt2* (AlUNC-4 binding sites deletion mutant) produced similar results (Fig. 4F). These results indicated that AlABD-B and AlUNC-4 are likely to be upstream regulators of *AlLuc1*.

EMSA was carried out to confirm the interaction between the recombinant AlABD-B and the motif sequence of the *AlLuc1* gene promoter in vitro. DNA-protein complex was detected, and its signal was significantly reduced in the presence of an unlabeled double-stranded oligonucleotide probe, corresponding to the motif sequence in the

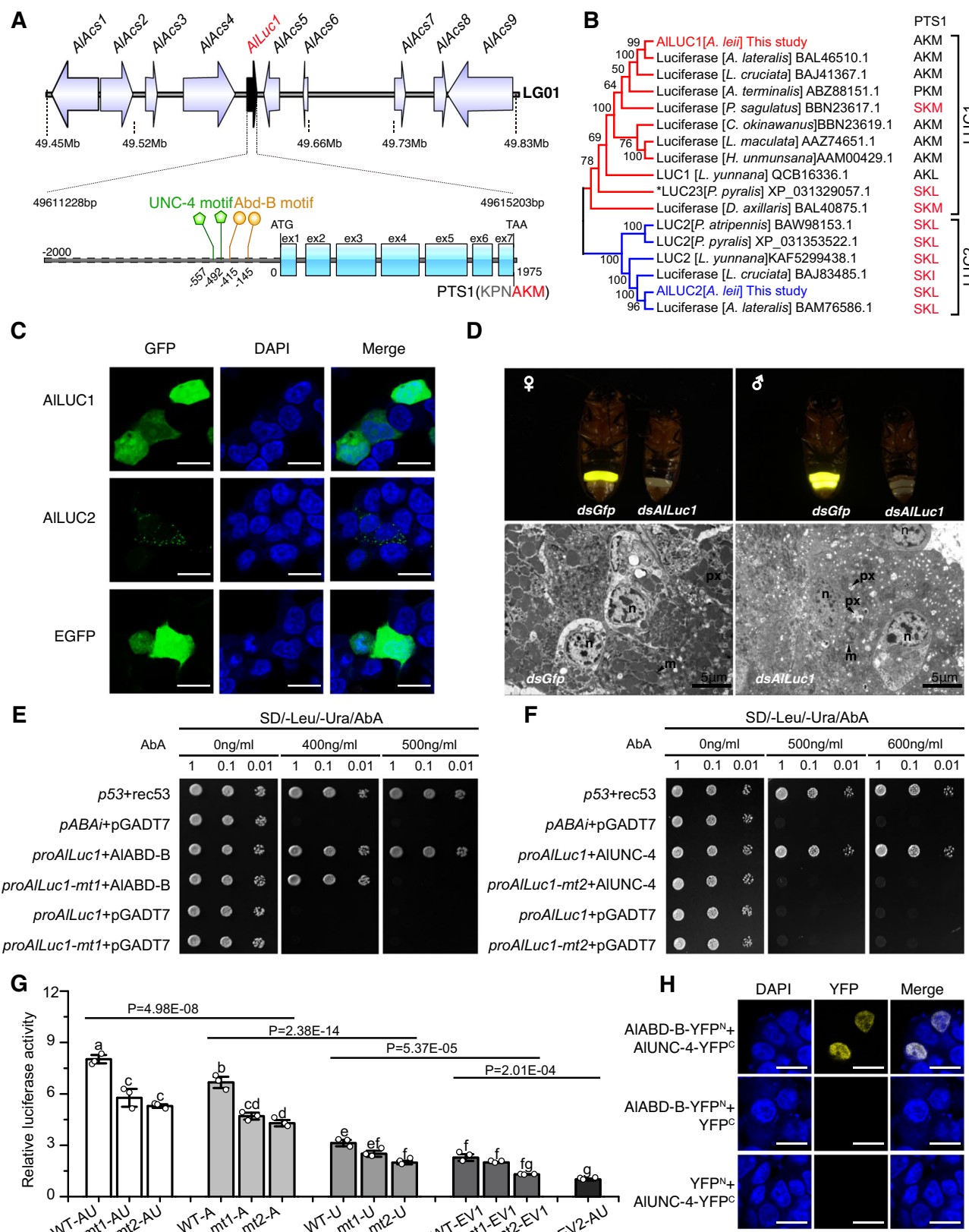

*AlLuc1* promoter (Supplementary Fig. 11A). To further detect the specificity of the interaction, we compared DNA-protein binding patterns of the mutation probe to the "wild-type" oligonucleotides (Supplementary Fig. 11B). The wild-type and mutation probe exhibited minor differences in sequence (Supplementary Fig. 11C), and striking discrepancy in binding patterns. No significant DNA-protein complex was detected.

Dual-luciferase reporter assays were carried out to conduct further validation of AlABD-B and AlUNC-4 transcriptional activation of the *AlLuc1* promoter. Different plasmids were constructed individually (Supplementary Fig. 12). The AlUNC-4 alone activated the reporter gene expression above the basal levels (Fig. 4G), but AlAbd-B activated the reporter significantly higher than AlUNC-4. However, a

**Fig. 4 | Bioinformatic analysis, subcellular localization, and functions of luciferase (LUC), and transcriptional regulation of *AlLuc1* by AlABD-B and AlUNC-4.**
**A** Genomic loci (top) and gene structure of *Alluc1*, comprising seven exons, are shown (bottom). Schematic diagram showing the two AlABD-B binding regions and two AlUNC-4 binding regions in the promoter of *AlLuc1*. **B** Evolutionary relationships of *Alluc1* and *Alluc2*. PTS1 of each sequence is shown next to its name, and PTS1 signals terminating with SK(L/I) are highlighted in red. **C** Subcellular localization of AlLUC1 and AlLUC2 in HEK293T cells (scale bar = 20 µm), n = 3 independent experiments, DAPI was used to stain the nucleus. **D** Functional verification of *AlLuc1* by RNAi (*dsAlluc1*). The results show that peroxisomes were cavitated (Scale bar = 5 µm), n = 5 independent experiments. n nucleus, px peroxisomes, m mitochondrion. **E** and **F** Detection of the interaction between *AlLuc1* promoter and

AlABD-B or AlUNC-4 using the yeast one-hybrid. Mutant promoters *proAlLuc1-mt1* or *proAlLuc1-mt2* were used as competitors to test the binding specificity. **G** Dual-luciferase assay in the 293T cell between AlABD-B, AlUNC-4, and *AlLuc1* promoter. WT, *proAlLuc1;* mt1, *proAlLuc1-mt1;* mt2, *proAlLuc1-mt2;* A, AlABD-B; U, AlUNC-4; EV1, empty vector pcDNA3.1; EV2, empty vector pGL4.17. All data are shown as the mean value of three replicates. Error bars represent standard deviation. Different lowercase letters above bars indicate significant differences as determined by the two-sided Student's *t*-test ($p < 0.05$; n = 3). The group data were analyzed using two-way ANOVA and multiple range tests. **H** Detection of the interaction between AlABD-B and AlUNC-4 using BiFC (Scale bar = 20 µm), n = 3 independent experiments. DNA: DAPI, blue; proteins: YFP, yellow. Source data are provided as a Source Data file.

combination of AlUNC-4 and AlAbd-B produced higher expression levels than either transcription factor alone, which suggests that AlUNC-4 enhanced the activity of the AlLuc1 gene promoter together with AlABD-B.

Cis-element mutants were also analyzed using the *proAlLuc1-mt1* (AlABD-B binding site mutant) and *proAlLuc1-mt2* (AlUNC-4 binding site mutant) (Fig. 4G). Each mutant was compared to a positive control (the wild-type plasmid with the relevant combinations of effector plasmids). A separate mutation of each cis-element abolished the additive activation (Fig. 4G, the reporters of *mt1* and *mt2* groups both significantly decreased compared with the wild-type group). However, according to the Dual-Luc results, mutations in promoters significantly reduced the interaction activity, but did not completely inhibit the expression of the reporter. The underlying reason for this may be that, even though high-scoring motif regions were deleted in *proAlLuc1-mt1* and *proAlLuc1-mt2* mutants, there might still exist low-scoring motif sequences within the promoter that have the potential to engage with AlABD-B or AlUNC-4.

The Dual-luciferase assay result demonstrated that AlABD-B and AlUNC-4 should bind DNA at their respective promoter cis-elements to attain an additive effect. We hypothesized that AlUNC-4 can combine with AlABD-B to enhance the expression activity of AlLuc1. Bimolecular fluorescence complementation (BiFC) was utilized to test this hypothesis. As shown in Fig. 4H, the expression of AlABD-B-YFP$^N$ or AlUNC-4-YFP$^C$ alone did not produce any fluorescence emission. However, co-expression of AlABD-B-YFP$^N$ and AlUNC-4-YFP$^C$ in HEK293T cells produced strong yellow fluorescence in the nucleus, indicating the presence of oligomers of AlABD-B and AlUNC-4.

In addition, through Western blot analysis, we found that the expression level of AlLUC1 protein significantly decreased in *dsAlAbd-B* and *dsAlUnc-4* groups compared with the wild-type and *dsGfp* groups (Supplementary Fig. 8B). Then, we used AF594-conjugated rabbit anti-LUC1 polyclonal antibody (red) as the primary antibody, and the cell nucleus was stained with DAPI (49,6-diamidino-2-phenylindole; blue). According to the immunofluorescence analysis, the red fluorescence representing the AlLUC1 protein was found in entire photocytes in the *dsGfp* group, whereas in the *dsAlAbd-B* group, the red fluorescence was weak and fluorescence signals discontinuous in photocytes, with many cavitations (Supplementary Fig. 8C). Overall, we conclude that both AlABD-B and AlUNC-4 are necessary for the AlLUC1 expression in light organs.

## Pex genes transported luciferase to peroxisome

Peroxisomes are simple organelles that consist of a protein-rich matrix surrounded by a single membrane. They are essential for cellular metabolism, including the β-oxidation of fatty acids, synthesis of etherlipid plasmalogens, and redox homeostasis. The membrane contains transporters, pores for solute transport, and proteins involved in diverse processes such as matrix and membrane protein sorting, organelle fission, and movement. Proteins that are required for peroxisome biogenesis are collectively called peroxins (PEX). Some peroxins play pivotal roles in the import of matrix proteins[28,29]. As we

have demonstrated that luciferase AlLUC1 functions within the peroxisomes (Fig. 4D), this suggests that AlLUC1 requires certain peroxins for import into peroxisomes. We performed RNAi analyses to verify whether the screened peroxins (AlPX11C.2, AlPXMP2, AlPEX5, AlPEX13, AlPEX14, AlPEX16, and AlPEX1) are involved in the import of AlLUC1. Transcript depletion of *AlPex13, AlPex14, AlPex5,* and *AlPxmp2* resulted in a significant decrease in the light intensity and a continuous glow compared with the control (Fig. 5A and Supplementary Fig. 13 and Supplementary Movie 3). The ultra-structure of the adult light organ in *dsAlPex13, dsAlPex14, dsAlPex5,* and *dsAlPxmp2* phenotypes revealed that some peroxisomes had shrunk or were empty (Fig. 5B). In addition, immunofluorescence analysis indicated that LUC1 antibody fluorescence signals were discontinuous in the dissected abdominal segment of *dsAlPex13* group (Supplementary Fig. 8C). Verification by the qPCR confirmed that the expression levels of *AlPex13, AlPex14, AlPex5,* and *AlPxmp2* were downregulated significantly in *dsAlPex13, dsAlPex14, dsAlPex5,* and *dsAlPxmp2* adults respectively. However, the expression level of *AlLuc1* remained unchanged (Fig. 5C, Supplementary Fig. 8B). Our results indicated that AlPEX13, AlPEX14, AlPEX5, and AlPXMP2 proteins are involved in the import of AlLUC1 into peroxisomes in *A. leii*.

Bioinformatics analysis showed that *AlPex13* and *AlLuc1* were located on chromosome LG01, while *AlPxmp2, AlPex5,* and *AlPex14* were located on chromosome LG05 (Fig. 5D). We carried out BiFC to verify pairwise interactions between AlLUC1, AlPEX13, AlPEX14, AlPEX5 and AlPXMP2. Co-expression of various gene combinations produced varying effects on fluorescence (Fig. 5E). Co-expression of four protein pairs in HEK293T cells produced strong yellow fluorescence: AlPEX13-YFP$^N$ and AlPEX14-YFP$^C$, AlPEX5-YFP$^N$ and AlPEX14-YFP$^C$, AlPEX5-YFP$^N$ and AlLUC1-YFP$^C$, and AlPEX14-YFP$^N$ and AlLUC1-YFP$^C$. These results indicate that these protein pairs interacted when expressed in HEK cells. Furthermore, the results suggested that AlPEX13 and AlPEX14 formed heterodimers and interacted with AlPEX5 - AlLUC1 in HEK cells.

To further confirm that AlPEX13 and AlPEX14 interacted to assist AlPEX5 in the import of AlLUC1 into peroxisomes, a multicolor bimolecular fluorescence complementation (mcBiFC) assay was carried out in HEK cells[30]. In this assay, the N-termini of the fluorescent proteins YFP (YN155) and CFP (CN155) were expressed with the C-terminus of CFP (CC155). When CC155 interacted with YN155, yellow fluorescence was produced, whereas when CC155 interacted with CN155, blue fluorescence was produced (Supplementary Fig. 14A). When pmCherry vector was fused with AlLUC1 and expressed in 293T cells, the red fluorescence representing the AlLUC1 protein could be found in both the nucleus and the cytoplasm. Following this, we fused AlPEX14 with CC155, AlPEX13 with YN155, and AlPEX5 with CN155 (Supplementary Fig. 14B), and co-expressed them together with mCherry-AlLUC1 in 293T cells. YFP and CFP fluorescence emissions revealed that AlPEX14 interacted with both AlPEX13 and AlPEX5 when expressed in HEK cells (Supplementary Fig. 14C). More importantly, we found changes in the subcellular localization of AlLUC1 protein in the co-expression group: most of the fluorescence was distributed in vesicular structures. Furthermore, in a few cells, we found that signals were distributed in a

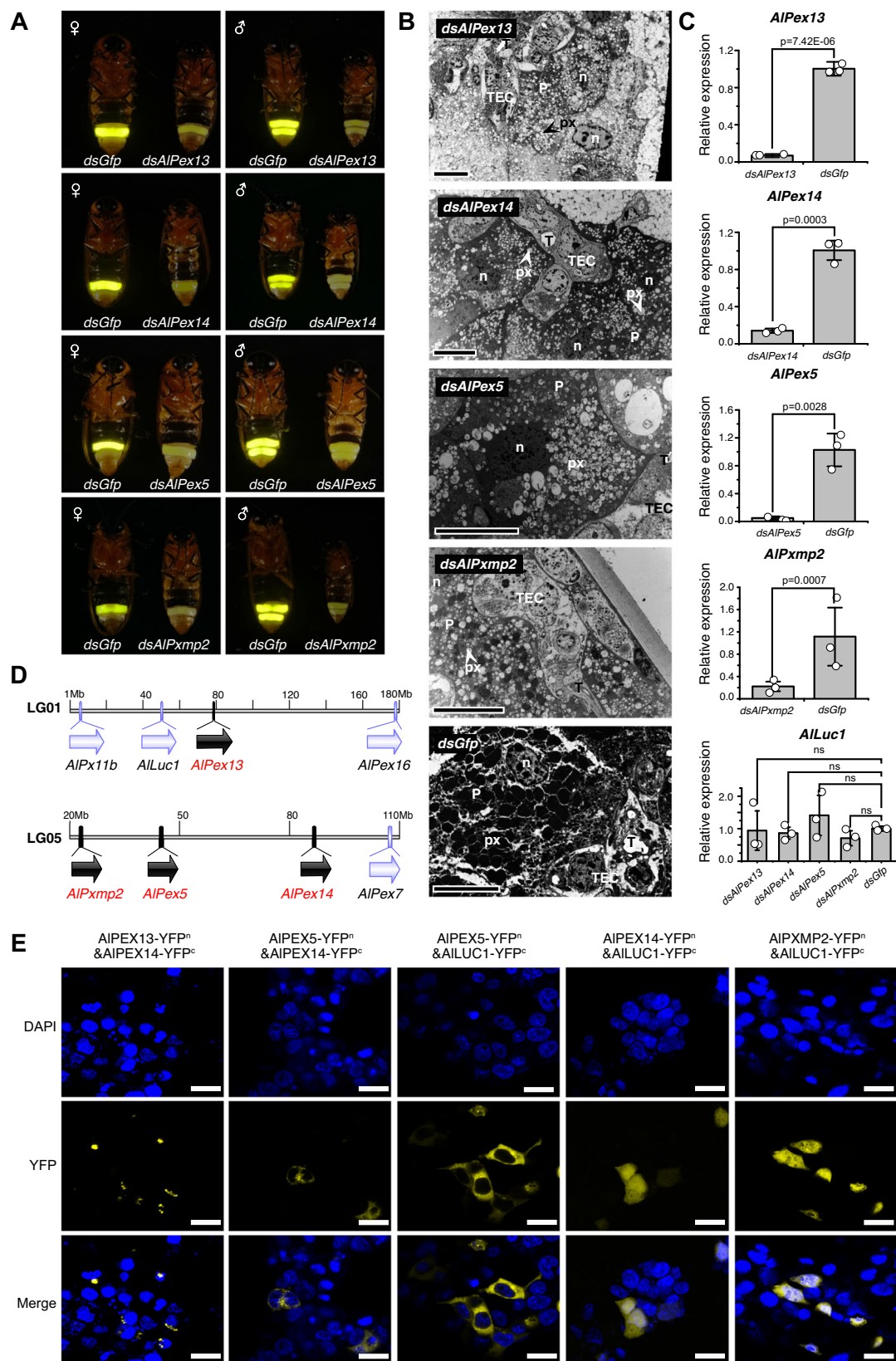

dotted pattern around the nucleus (Supplementary Fig. 14C, the bottom panel). We hypothesize that this might be the peroxisome.

Our research on interactions between AlPEX13, AlPEX14, AlPEX5, and AlLUC1 is consistent with the known mammalian peroxisomal protein import machinery. PEX14 and PEX13 are well-known constituents of the human docking complex, which is related to the peroxisomal protein import machinery. PEX5 binds to cargo proteins containing a PTS1 peroxisomal targeting signal in the cytosol and translocates them into the peroxisome matrix by passing through the PEX13-PEX14 docking complex along with cargo proteins[31,32].

In addition, co-expression of AlPXMP2-YFP$^N$ and AlLUC1-YFP$^C$ in HEK293T cells also produced strong yellow fluorescence (Fig. 5E), so

**Fig. 5 | Verification of *Pex genes* involved in the import of AlLUC1 into the peroxisome. A** Dim and continuous glow phenotypes caused by the knockdown of *Alpex13*, *Alpex14*, *Alpex5*, and *AlPxmp*2. **B** The ultra-structure of adult light organs in experimental and control phenotypes. *n* = 5 independent experiments. T trachea, TEC tracheal end cells. P photocyte, n nucleus, px peroxisomes (Scale bar, 10 μm). **C** The relative expression levels of genes in *dsAlpex13*, *dsAlpex14*, *dsAlpex5*, and *dsAlPxmp*2 treatment groups inferred by qPCR. The data are presented as mean values ± SD of three biological replicates (2 males and 1 female). The data were

analyzed using one-way ANOVA and Tukey's multiple range tests, where ns indicates a nonsignificant result ($p > 0.05$). **D** A schematic of the arrangement of *AlPex13*, *AlPex14*, *AlPex5*, and *AlPexmp2* genes on the chromosomes (black arrows). Genes are shown in the order in which they are found on the chromosomes but, for clarity, only peroxin genes are shown. The positions of other peroxin genes and AlLuc1 are shown using gray arrows. **E** Protein interaction analysis by BiFC in the 293T cells. DAPI was used to stain the nucleus. *n* = 3 independent experiments (scale bar = 20 μm). Source data are provided as a Source Data file.

we hypothesize that it may be involved in the identification or transportation of AlLUC1.

## Discussion

Uncovering the mechanisms underlying the evolution of novel complex traits is a central challenge in biology[9]. When a new feature appears in a species, but ancestral states do not exhibit obvious homology to the new trait, the commonly applied research strategy to solve this scientific problem is to carry out genomic evolutionary analyses, such as the evolution of genome size, and expansion and contraction of gene content. Previously, multiple studies have relied on comparative genomic approaches to explain the evolution of novel complex traits, such as the genome evolution during the adaptive radiation of Heliconiini butterflies[33], and adaptation to urban environments and developmental plasticity of the American cockroach[34].

The light organs of fireflies are complex traits that lack even remote homology to structures outside the luminescent beetle families[9]. Previous studies relied on comparative genomic methods to explore the origin and evolution of luciferases and bioluminescence in beetles[23]. Through genome sequencing and multi-omics analysis of two fireflies *L. yunnana* and *A. terminalis*, an enrichment luciferin synthesis pathway was proposed, and the convergent evolution of bioluminescence in insects was confirmed[10]. However, the genetic regulation of light organ development remained unstudied until now. Similarly, molecular mechanisms producing structural differences of luminous organs between different firefly species also remain unknown. After assembling the genome of *A. leii*, we conducted a genomic evolutionary analysis of 12 species (Fig. 1E): GO and KEGG enrichment analyses revealed that expanded gene families in *A. leii* are associated with the apoptosis pathway, insect hormone biosynthesis pathway and pathogenic infection pathway (Supplementary Fig. 15). They also comprised the P450 family, so we hypothesize that these features might be associated with chemical defenses and the unique lifestyle of fireflies, on the basis of the P450 gene family research of Fallon et al.[23]. However, the above steps, genome assembly and comparative analysis of firefly genomes, revealed that the identification of gene families that underwent expansions is insufficient to unveil the molecular mechanisms of light organ development in fireflies. To address this issue, further research on the acquisition of new functions of known genes might be necessary.

Homeobox genes act at many levels within developmental gene hierarchies: at the "executive" level they regulate genes that in turn regulate large networks of other genes (like the gene pathway that forms an appendage). They also directly regulate the so-called realisator genes or effector genes, which act at the bottom of such hierarchies to ultimately form tissues, structures, and organs of each segment[35].

Adult light organs are complex, integrated structures, nearly always restricted to sixth or seventh abdominal segments. Their flashes or glow play a well-understood role in courtship rituals among conspecific adult fireflies. RNAi of the Abdominal-B in fireflies results in extensive disruption of the adult light organ and non-luminescence[9]. Their unique structure and function led us to propose two main hypotheses. The first main hypothesis posits that the origin of the light organ was enabled by the evolution of a novel regulatory role for the

homeobox genes related to primitive segment development. The second main hypothesis posits that the new function of homeobox genes is related to neuronal development, thus directly or indirectly regulating the flash behavior.

Through screening homeobox genes related to the adult light organ development, we finally selected eight homeobox genes to conduct functional studies using RNAi methodology. Among the eight genes, *Abd-A*, *Ubx*, *Abd-B*, *Antp*, and *Ap* play vital roles in the primitive segment development in model insects, while *Unc-4*, *Repo*, and *Shox2* are related to neural development. Abdominal-A (gene name is *Abd-A*) is essential for the formation of embryonic and abdominal segments in insects[36], Ubx ultrabithorax (*Ubx*) acts as a genetic switch to modify specific morphological features in thoracic region in insects[37], and Abdominal-B (*Abd-B*) is required for the proper development of the posterior abdomen through suppression of expression of *Abd-A*[38]. Antennapedia (*Antp*) is essential for wing development in insects[39], while apterous (*Ap*) is required for the normal development of the wing, halters imaginal disks[40], and functions to control the neuronal pathway selection[41]. Homeobox protein UNC-4 regulates synaptic specificity in insects[42], while reversed polarity protein (*Repo*) is required for the acquisition of glial fate and subsequent terminal glial differentiation[43]. Finally, short-stature homeobox protein 2 (*Shox2*) may be a growth regulator and have a role in specifying neural systems involved in processing somatosensory information[44]. The two types of genes correspond to the two main hypotheses.

However, during the functional validation, we discovered results that were not entirely consistent with our hypotheses. We found that homeobox family genes were required for the development of the adult light organ in fireflies. Specifically, AlABD-B emerged as the decisive transcription factor for normal light organ development, whereas AlABD-A, which typically determines segment fate in other insects, only regulated V3-5 epidermal pigment deposition. Furthermore, AlANTP was proven to be associated with flash control, and AlUNC-4 was involved in the development of the adult light organ.

Furthermore, Abd-B is transcribed in two isoforms in *D. melanogaster*: a regulatory protein (Abd-BR), and a morphogenic protein (Abd-BM). Abd-BR suppresses embryonic ventral epidermal structures in the eighth and ninth segments of the *Drosophila* abdomen, and both of them are involved in the development of the tail segment[45]. On the contrary, no isoform of *AlAbd-B* was found in the genomic and transcriptomic data of *A. leii*.

These differences in genetic regulations and gene functions in fireflies suggest that the molecular regulatory mechanisms known from model insects may not be suitable for firefly research. Many known genes in fireflies could potentially have new functions, underpinning the evolutionary emergence of new traits and providing fireflies with a completely new evolutionary potential. This highlights the need to functionally confirm and if needed redefine many known genes in fireflies.

In several instances, multiple Hox genes are known to be expressed within the same segment;[46] in such cases, posterior Hox genes suppress the function of comparatively anterior Hox genes, commonly by repressing their transcription. This phenomenon is known as posterior dominance[47]. Knockouts of Abd-B and Abd-A induced homeotic transformations that violated the "posterior

dominance" principle[48], This can be logically explained via a combinatorial "Hox code" model[49,50]. Based on this model, we propose a hypothesis that the combined expression of multiple homeobox genes would have a combinatorial influence on downstream targets in the light organ of adult fireflies. Therefore, it is important to study the interactions between Abd-B and other transcription factors in future.

In this study, we discovered the interaction between transcription factors AlABD-B and AlUNC-4 in the light organs of firefly *A. leii*. Interactions between transcription factors DmUNC-4, DmABD-A, and DmUBX were previously found in *Drosophila melanogaster* using the BiFC method[51], but until now there has been no evidence of the interaction between UNC-4 and ABD-B. Here we showed that after *dsAlAbd-B*, the adult light organs, commonly consisting of three layers (epidermis (EP), photogenic layer (PL), and dorsal layer (DL)), exhibited irregular morphology and were not luminescent. After *dsAlUnc-4*, the structure of the PL layer was significantly changed in adult light organs (accompanied by non-luminescence), while EP and DL layers did not exhibit major changes. Hence, we proposed a theory about the mechanism of the development of the light organs (Supplementary Fig. 16). AlABD-B regulates the development of the entire adult light organ, and its RNAi results in a darkened cuticle, disruption of the bioluminescent layer's peroxisomes, and loss of the DL layer. AlUNC-4 participates in the development of the light organs and regulates the development of peroxisomes in photocytes by interacting with AlABD-B. Reduced expression of AlABD-B or AlUNC-4 leads to the malformation of peroxisomes, a reduced number of peroxisomes, mislocation of the matrix proteins of peroxisome (such as AlLUC1, etc.), and interruption of the normal light emission by light organs.

In this study, the interactions between AlABD-A and AlABD-B were not studied because the *AlAbd-A* transcript exhibited a downward expression trend during the development of the light organ (Supplementary Fig. 5) and *dsAlAbd-A* treatment did not produce a significant difference in the light intensity and flash rate (Fig. 2A). The epidermis of the light organ lacks pigmentation, allowing light to pass through. Furthermore, the epidermal layer is closely connected to the photogenic layer (Fig. 2C), which indicates that the two layers may share a common regulatory model. Both *Abd-A* and *Abd-B* are known to influence the pigmentation of abdominal cuticle in model insects. In future research, we will focus on the synergistic regulatory relationship between AlABD-B and AlABD-A by studying differences in the molecular regulation of abdominal cuticle pigmentation in ventrites 3–7, including the light organ.

Our results showed that the light organ was nearly non-luminescent when *AlLuc1* expression was interfered with, or when AlLUC1 was translated normally but it could not be targeted to the peroxisome matrix. Therefore, it is important to study the process of AlLUC1 entering the peroxisomes.

PEX14 and PEX13 are recognized constituents of the human docking complex, which is related to the peroxisomal protein import machinery[52]. PEX5 binds to cargo proteins containing a PTS1 peroxisomal targeting signal in the cytosol, and translocates them into the peroxisome matrix by passing through the PEX13-PEX14 docking complex along with cargo proteins[31,32]. Our analyses revealed that AlPEX13 and AlPEX14 interacted to assist AlPEX5 in the import of AlLUC1 into peroxisomes. However, during the subcellular localization testing of the AlLUC1 protein in 293T cells, we found a problem: AlLUC1 was located both in the cytoplasm and nucleus. These results are not consistent with previous research[4,15,25]. LUC1 of *P. pyralis* (LUC23, NCBI registration number XP_031329057.1) was targeted to peroxisomes in mammalian cells[15]. Peroxisomal matrix proteins rely on two types of peroxisomal targeting signals (PTS) located at the C-terminus (PTS1) and N-terminus (PTS2). PTS1 is carried by most peroxisomal matrix proteins, and it is marked by the tripeptide -SKL, or conserved variants thereof. Our analysis of PTS1 motifs at the C-termini of all known luciferases revealed that all LUC2 PTS1 signals consistently

terminated with SK(L/I), whereas LUC1 PTS1 signals were more variable: most terminated with AKM, and a few with (S/A/P)K(L/M) (Fig. 4B). Research on PTS1 signals in plants showed that AKM is a low-frequency PTS1 signal; its efficiency in targeting the peroxisome depended on the three residues immediately upstream of the tripeptide[53]. AlLUC1 PTS1 signal motifs ended with AKM, and the three residues immediately upstream of the tripeptide were KPN (Fig. 4A), which did not match the basic-nonpolar-basic pattern and may not enhance AKM-type signals targeted to the peroxisome. As AlLUC1 with low-frequency PTS1 tripeptides cannot be targeted to peroxisomes by peroxin proteins in mammalian cells, our analyses indicate that AlLUC1 of *A. leii* requires specific transport proteins to target it to the peroxisome.

McBiFC helped us to detect the roles of AlPEX13, AlPEX14, and AlPEX5 in the import of AlLUC1 into peroxisomes in HEK cells. Most of the fluorescence signals were observed in vesicles, and a few were observed in the peroxisome. This indicates that LUC1 depends on vesicular transport into the peroxisome matrix. In order to deepen our understanding of AlLUC1 transport, we will conduct research on the targeted intracellular vesicle transport in the future.

PXMP2 is a widely expressed and abundant homotrimeric peroxisomal membrane protein (PMP) that displays channel-forming activity and allows free diffusion of compounds with an upper molecular size limit of 300–600 Da. Furthermore, PXMP2 was the first peroxisomal channel identified in the mammalian peroxisomal membrane, which led to the prediction that the mammalian peroxisomal membrane is permeable to small solutes, while the transfer of "bulky" metabolites, e.g., cofactors (NAD/H, NADP/H, and CoA) and ATP, requires specific transporters[54]. However, in our research, AlPXMP2 interacted with AlLUC1 and it may be involved in the identification or transportation of AlLUC1. This result contradicts the "small molecule channels" hypothesis. Lismont's research showed that loss of PEX11B, a peroxisomal membrane-shaping protein, facilitates the permeation of molecules up to 400 Da, decreases peroxisome density in HEK-293 cells, and causes partial localization of peroxisomal matrix proteins to mitochondria[19]. PEX11B and PXMP2 have similar structures and functions, so this finding may bear relevance for our findings. Our research reveals a role for AlPXMP2 in the targeting of AlLUC1 protein to peroxisomes in *A. leii*.

Compared with other model insects, firefly breeding is relatively difficult, which limits our research. For example, transcriptomic data from different developmental stages are available for males, but not for females. In the follow-up study, we will strengthen the research on the differences between the two sexes in fireflies.

In conclusion, our study revealed a series of novel mechanisms by which two key homeobox transcription factors regulate the development of adult light organs and bioluminescence. We identified two key homeobox transcription factors: AlABD-B regulated AlUNC-4, and they interacted with each other. Peroxins *AlPex13* and *AlPex14* were regulated by AlABD-B. *AlLuc1*, *AlPex5*, and *AlPxmp2* were regulated by AlABD-B and AlUNC-4 simultaneously, but AlABD-B was essential. AlUNC-4 enhanced the transactivation activity of AlABD-B, thus strongly activating the expression of downstream genes. AlPEX13 and AlPEX14 interacted to assist AlPEX5 in the import of AlLUC1 into peroxisomes. AlPXMP2 and AlLUC1 interacted and were involved in the import of AlLUC1 into peroxisomes (Fig. 6).

## Methods
### Animals
Larvae, pupae, and adults were obtained from the aquatic firefly *Aquatica leii* breeding lab (a lab established in Wuhan City, Hubei Province, solely to breed *A. leii* from the original firefly population collected from Hangzhou City, Zhejiang Province), and kept in the laboratory at $25 \pm 1\,°C$ for 24 h under $70 \pm 5\%$ humidity and a 14:10 h light/dark (L:D) photoperiod. All experiments were performed after

anesthetization with $CO_2$. For treatments, three biological replicates were performed, with 5 fireflies in each replicate. The same 1-day-old adult (after emergence from the pupae stage) male specimen was used for both MGI and ONT genome sequencing. Another 1-day-old adult male specimen was used for Hi-C sequencing. Twenty 5th-instar larvae were used for karyotype observation. Tissues corresponding to the adult light organs (ventrites 6 and 7, abdomen cuticle) were dissected from male pupae specimens at different developmental stages (1d-pupae, 3d-pupae, and 5d-pupae) for RNA-seq.

The animal experiments were performed according to the procedures approved by the Laboratory Animal Welfare and Ethics Committee of Huazhong Agricultural University.

### Genome assembly using third-generation long reads

High molecular weight genomic DNA was prepared by the SDS (sodium dodecyl sulfate) extraction method followed by purification with a QIAGEN® Genomic kit according to the standard operating procedure provided by the manufacturer. Genomic DNA (1–1.5 µg) was randomly fragmented by Covaris, and the fragmented DNA was selected to an average size of 200–400 bp using the Agencourt AMPure XP-Medium kit. The selected fragments were circularized and sequenced on an MGISEQ2000 platform (GrandOmics, Wuhan, China). Quality-filtered reads were subjected to a 17-mer frequency distribution analysis using the KMC (v3.2.1)[55] program. The genome size of *A. leii* was estimated using FindGSE[56] and GenomeScope (v1.0.0)[57]. The heterozygosity and repeat content of *A. leii* genome were estimated by combining the simulation data results for

*Arabidopsis thaliana* with different heterozygosity levels and the frequency peak distribution of 17 kmer.

ONT library preparations were conducted using 2 µg of DNA per sample, and sequencing was then performed on a Nanopore PromethION sequencer (Oxford Nanopore Technologies) at the Genome Center of Grandomics (Wuhan, China). NextDenovo (v2.3.0) was used for de novo genome assembly. To improve the accuracy of the assembly, contigs were refined with Racon (v1.3.1) using ONT long reads and Nextpolish (v1.2.4) using MGI short reads with default parameters.

The coverage of expressed genes of the assembly was examined by aligning all RNA-seq reads against the assembly using HISAT2 (v2.1.0)[58] with default parameters. To avoid including mitochondrial sequences in the assembly, the draft genome assembly was submitted to the NT library and aligned sequences were eliminated.

### Chromosome number, karyotype, and chromosome assembly using Hi-C data

Chromosomal preparations were obtained from the gonads of twenty 5th-instar larvae. The gonads were surgically removed, placed in a 10 mg/mL colchicine solution (insect saline solution) for 120 min, and then subjected to a hypotonic treatment for 30 min. All gonads were fixed in Carnoy I (three parts methanol: one part acetic acid) for 60 min. For the preparation of slides, the gonads were macerated in 45% acetic acid until a cell suspension was acquired, which was spread over a slide and dried on a metal plate at 40 °C. Chromosomal preparations were stained with a 3% Giemsa solution for 10 min. Light

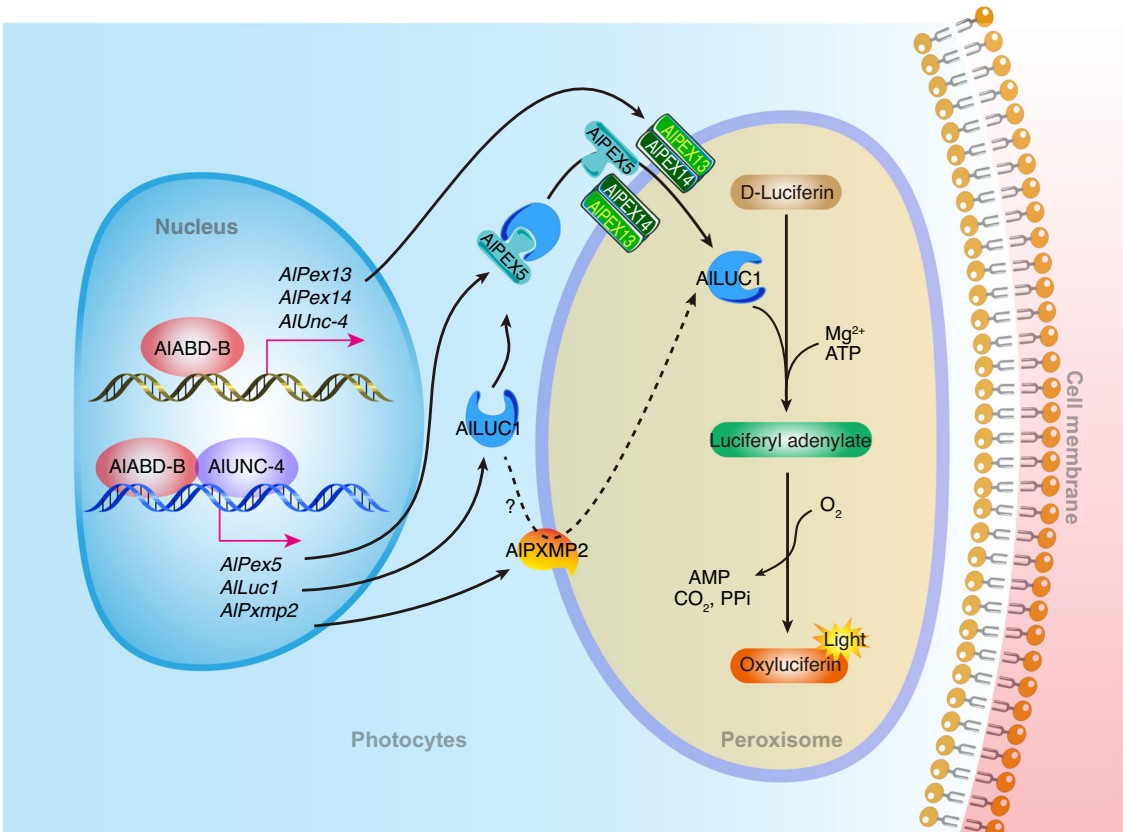

**Fig. 6 | The pathways through which key homeobox transcription factors AlABD-B and AlUNC-4 regulate the development of adult light organs and bioluminescence in the firefly *A. leii*.** *AlUnc-4, AlPex13,* and *AlPex14* were regulated by AlABD-B. AlABD-B interacted with AlUNC-4, both of them regulated *AlLuc1, AlPex5,* and *AlPxmp2*. AlPEX13 and AlPEX14 interacted to assist AlPEX5 in the import of AlLUC1 into peroxisomes. AlPXMP2 and AlLUC1 interacted and were involved in the import of AlLUC1 into peroxisomes. AlLUC1 catalyzed the oxidation of firefly luciferin with molecular oxygen in the presence of ATP and $Mg^{2+}$ to emit yellow-green light.

microscope photomicrography was carried out using a Zeiss photomicroscope.

The whole fresh *A. leii* body was vacuum infiltrated into the nuclei isolation buffer supplemented with 2% formaldehyde. The fixed tissue was then ground to powder before re-suspending it in the nuclei isolation buffer to obtain a suspension of nuclei. The purified nuclei were digested with 100 units of *DpnII* and marked by incubation with biotin-14-dCTP. The ligated DNA was sheared into 300–600 bp fragments, blunt-end repaired, and A-tailed, followed by purification through biotin-streptavidin-mediated pull-down. Finally, the Hi-C libraries were quantified and sequenced using the Illumina Novaseq platform (GrandOmics, Wuhan, China). Clean paired-end reads were mapped to the draft assembled sequence using bowtie2 (v2.3.2) (end-to-end --very-sensitive -L 30) to obtain unique mapped paired-end reads. The scaffolds were further clustered, ordered, and oriented onto chromosomes using LACHESIS (https://github.com/shendurelab/LACHESIS). To check the completeness and quality of the assembly, BUSCO (v5.1.3)[59] was used to search the 1367 bench-marking universal single-copy orthologous genes in the insecta_odb10 database.

## RNA-seq

For gene annotation, we sequenced the transcriptomes of male pupal light organs (cuticle of ventrites 6 and 7) collected from three different developmental stages: early pupal stage (1-day-old, Lo_E), middle pupal stage (3-day-old, Lo_M), and late pupal stage (5-day-old, Lo_L). We collected three biological replicates per experimental group, with 15 specimens per replicate. Total RNA was extracted from these samples using TRIzol Reagent (Invitrogen). The RNA concentration was determined on a Nano-Drop 2000 spectrophotometer (Thermo Scientific), and further checked by 1.5% agarose gel electrophoresis. PolyA RNA-sequencing (RNA-seq) libraries with an insert size of 150 bp were prepared and sequenced on the Illumina Hiseq platform (BGI, Shenzhen, China). Raw sequences were filtered using Trimmomatic (v0.39)[60]. A total of ~6 Gb of clean data was obtained for each sample. Differential gene expression was measured following the Hisat2-stringtie pipeline[61]. First, clean reads were mapped against the assembled *A. leii* genome using hisat2 (v2.1.0)[58]. Then, the expected TPM was calculated using Stringtie (v1.3.4) software for each individual. Finally, DEGs were identified by DESeq2 (v1.40.2) with fold change >2 and $P < 0.05$[62].

## Genome annotation

Tandem repeats were identified using the software programs GMATA (v2.2)[63] and Tandem Repeats Finder (TRF v4.07b)[64]. While GMATA identifies only simple repeat sequences (SSRs), TRF recognizes all tandem repeat elements in the whole genome.

Three independent approaches, including ab initio prediction using software AUGUSTUS (v3.3.1)[65], homology search using software GeMoMa (v1.6.1)[66], and reference-guided transcriptome assembly by software PASA(v2.3.3)[67], were used for gene prediction in a repeat-masked genome. Functional annotation of genes, and motif and protein domain prediction, were conducted via Blastp (v2.7.1) comparisons with data in the following public databases: SwissProt, NR, KEGG, KOG, and Gene Ontology. Transfer RNAs (tRNAs) were predicted using tRNAscan-SE (v2.0) with "eukaryote" parameters. MicroRNA, rRNA, small nuclear RNA, and small nucleolar RNA were detected using Infernal (v1.1.2) cmscan[68] to search the Rfam database. rRNAs and their subunits were predicted using RNAmmer (v1.2)[69].

## Orthology and phylogenomics

A total of 12 representative insect species, including *A. leii*, were selected for orthology analysis (their protein sets are listed in Supplementary Table 7). The downloaded protein sequences were aligned with *A. leii* genome data using OrthMCL(v2.5.4)[70]. Firstly, protein sets were identified for all 12 species and the longest transcript of each gene

was extracted while discarding miscoded genes and genes exhibiting premature termination codons. The extracted protein sequences were then aligned pairwise to identify conserved orthologs using Blastp (v2.7.1) with an E-value threshold of 1E-5. Orthologous inter-genome gene pairs, paralogous intra-genome gene pairs, and single-copy gene pairs were further identified using OrthMCL. The *A. leii* proteins with no homologs in the other 11 insect genomes were extracted as species-specific genes, including the *A. leii*-specific unique genes and unclustered genes. Functional annotation of species-specific genes and enrichment tests were performed using information from homologs in the Gene Ontology (http://www.geneontology.org/) and KEGG (Kyoto Encyclopedia of Genes and Genomes, https://www.kegg.jp/kegg/) databases[71].

On the basis of the orthologous gene sets identified with OrthMCL, molecular phylogenetic analysis was performed using the shared single-copy genes. Briefly, the coding sequences were extracted from the single-copy families and each set of orthologs was aligned using MUSCLE (v3.8.31)[72]. The resulting alignments were processed using Gblocks (v0.91b)[73,] with parameters designed to maximize the retention of informative sites[74]. The alignments were concatenated, converted to the Phylip format, and an appropriate model of amino acid substitution was selected using ProtTest (v3.0)[75]. Using the selected LG+I+G model, phylogenetic trees were generated using PhyML (v3.0)[76] with 1000 bootstrap replicates. The generated tree file was visualized using Figtree (v1.4.3). Based on the phylogenetic tree, the RelTime of MEGA-CC (v10.1.8)[77] was utilized to compute the mean substitution rates along each branch and estimate the species divergence time. Two fossil calibration times were obtained from the TimeTree database (http://www.timetree.org/) as the time control, including the divergence times of *D. melanogaster* and *T. castaneum* (estimated at 308 MYA, range: 234–370 MYA), and *I. luminosus* ad *P. pyralis* (133 MYA, range: 103–127 MYA).

## Identification of the homeobox gene family

Transcription factors (TF) comprise a group of protein molecules that can specifically bind to a specific sequence upstream of the 5′ end of a gene, thereby ensuring that the target gene is expressed at a specific time and space with a specific intensity. For *A. leii* TFs, we compared all *A. leii* proteins with the AnimalTFDB (v3.0) database[78] and obtained the corresponding TF family.

We separately built a local blast database by downloading the homeobox domain model (PF00046) and homeobox_KN domain model (PF05920) from the Pfam database (http://pfam.xfam.org/) and used it as a BLAST query. Then, a search was performed using HMMER (v3.0) (http://hmmer.org/). Additionally, 105 *Tribolium castaneum* homeobox protein sequences were downloaded from the HomeoDB (http://homeodb.zoo.ox.ac.uk, accessed on 25th February 2022) database[14]. For queries, we performed Blastp (v2.7.1) searches of the *A. leii* genome with an E-value threshold of 1E-10. Finally, the results obtained from HMMER and Blastp were searched using SMART (v9.0) (http://smart.embl-heidelberg.de/) to make structural predictions. Protein sequences lacking the homeobox domain and non-full-length genes were rejected. All full-length homeobox protein sequences of *A. leii* and *T. castaneum* were aligned using MAFFT (v7.5)[79]. Gap sites were removed with trimAl (v1.2)[80] using the "-automated1" command. Then, IQ-TREE (v2.1.2)[81] was used to construct a maximum-likelihood (ML) phylogenetic tree. The best-fit substitution model LG+G4 was determined using ModelFinder (part of IQ-TREE version 1.6.1)[82]. The bootstrap values were 20000. Homeobox genes were renamed according to sequence homology with *T. castaneum*.

## DNA cloning and bioinformatics analyses

CDS sequences of *AlAbd-A, AlAbd-B, AlAntp, AlUnc-4, AlUbx, AlShox2, AlRepo, AlAp2, AlLuc1, AlLuc2, AlPex1, AlPex5, AlPex13, AlPex14, AlPex16, AlPxmp2, AlPx11c.1,* and *AlPx11c.2* were cloned using the primers shown

in Supplementary Data 3. The amplified CDS sequences were cloned into the pMD18-T Simple vector (Takara) and then transformed into *E. coli* DH5a Competent Cells (Bio-Transduction Lab Co.Ltd). The plasmid was extracted, verified using Sanger sequencing, and subsequently used for dsRNA synthesis and expression vector construction. cDNA and its protein sequence were predicted using DNAMAN (v7.0.2), and PST1 signal peptides of luciferases were predicted using PSORT Prediction (http://psort1.hgc.jp/form.html). The evolutionary history of *AlLuc1* and *AlLuc2* was inferred using the Neighbor-Joining method implemented in MEGA7 (7.0.21)[83]. The percentages of replicate trees in which the associated taxa clustered together in the bootstrap test (1000 replicates) are shown next to the branches.

### RNAi in vivo
For RNA interference (RNAi) experiments, double-stranded RNA (dsRNA) of *dsAlAbd-A*, *dsAlAbd-B*, *dsAlAntp*, *dsAlUnc-4*, *dsAlUbx*, *dsAl-Shox2*, *dsAlRepo*, *dsAlAp2*, *dsAlLuc1*, *dsAlLuc2*, *dsAlPex1*, *dsAlPex5*, *dsAlPex13*, *dsAlPex14*, *dsAlPex16*, *dsAlPxmp2*, *dsAlPx11c.1*, and *dsAlPx11c.2* were produced by in vitro transcription using the TranscriptAid T7 High Yield Transcription Kit (Thermo Scientific) according to the manufacturer's protocol. The primers used for dsRNA synthesis are shown in Supplementary Data 3. The green fluorescent protein (GFP) gene was used as the negative control (*dsGfp*). Both male and female one-day-old pupae were injected with 2 μg of dsRNA (total injection volume 500 nL per insect) through the abdominal segment membrane, between V6 and V7. Injections were delivered via a WPI PicoPump pressure delivery system (504127) with a WPI smart touch. Injected pupae were placed in a small box containing moist filter paper until the emergence of adults.

### RNA sequencing and differential gene expression analysis
The light organs of 2-day-old adult male fireflies that underwent *AlAbd-B* RNAi, and the whole bodies of 2-day-old adult males that were injected with *dsAlUnc-4* in vivo were used for total RNA extraction. We collected three biological replicates per experimental group, with 15 individuals in each replicate. The total RNA of the *dsGfp* treatment group was extracted as a control, and then RNA sequencing was conducted according to the methodology described above (RNA-seq section). Clean RNA-seq reads were aligned to the *A. leii* genome using HISAT2 v2.1.0[58], and RSEM was used to calculate the gene expression levels for each sample[84]. DEseq2 (v1.40.2)[62] was used to moderate the estimation of fold change and dispersion for RNA-seq data.

### Quantitative real-time PCR
For each dsRNA treatment group, we sampled 2 males and 1 female. The whole bodies of 2-day-old adults were collected, and the total RNA was extracted using the RN07 EASYspin RNA Mini Kit (Aidlab). One μg of RNA was used for cDNA synthesis with PrimeScript™ RT reagent Kit with gDNA Eraser (Perfect Real Time) (TaKaRa). Real-time quantitative PCR was performed using TB Green Premix Ex Taq II (Tli RNaseH Plus) (TaKaRa). Primers were designed using the Primer3 web tool (https://primer3.ut.ee/)[85]. The primer pair names and sequences are shown in Supplementary Data 3. Primer sequences were screened using a BLAST search to confirm specificity, and the PCR products were run on an agarose gel to confirm that products of the expected size were detected. qRT-PCR analysis was conducted using a CFX Connect Real-time PCR Detection System (Bio-Rad, Hercules, CA). The reactions were prepared as follows: 5 μL of 20-fold diluted cDNA template, 12.5 μL 2 × SYBR Green PCR Mix, 0.5 μM of each gene-specific primer, and ddH$_2$O for the remaining volume. The PCR program consisted of an initial denaturation at 95 °C for 3 min, followed by 40 repeated cycles, each consisting of 95 °C for 50 s, 60 °C for 30 s, and plate reading. The melting curve covered 65 °C–95 °C, with increments of 0.5 °C for 5 s, and then the plate read. Gene expression was calculated

using the $2^{-\Delta\Delta Ct}$ method[86] and it was normalized to the abundance of the reference gene *AlEf1a*[87].

### Sectioning and ultrastructural observations
Adult light organs from the RNAi and the control groups were carefully dissected and fixed in 2% cold glutaraldehyde, postfixed by adding 2% cold osmiumtetroxide into the same buffer, dehydrated through a graded acetone series, and embedded in Araldite. Semi-thin sections for light microscopy were stained with Delafield's haematoxylin and eosin, photomicrography was carried out using a Zeiss photomicroscope. Ultrathin sections for the transmission electron microscopy were double-stained with uranyl acetate and lead citrate for a few minutes each[88]. Finally, they were examined using a H7650 TEM microscope (Hitachi, Tokyo, Japan) at an accelerating voltage of 75 KV.

### Flash rate and relative light intensity
Two-day-old adults were photographed and filmed using a stereo microscope coupled with a video camera (Sony A7s2). Photographic parameters were: ISO 4000 and an exposure time of 2 s. The video parameters were ISO 102000 and a shutter speed of 1/60. Videos were recorded in the mp4 format (4K) and we used software developed specifically for the analysis (output of flash signals rendered as relative intensities against time)[89]. The maximum relative light intensity was recorded and compared between the experimental and control phenotypes ($n = 16$)[90] The recorded flash patterns of male adults are presented in Supplementary Movies 1–3.

### Detection of luciferin content by HPLC
To detect the content of luciferin in tissues, we used HPLC analyses as described previously[91]. The specimens were frozen and ground in liquid nitrogen, and then homogenized with ddH$_2$O at 95 °C for 10 min. After centrifugation, the supernatant was washed with an equal volume of n-hexane, and the aqueous layer was subjected to centrifugal filtration with Ultrafree-MC (PVDF, 0.45 mm; Millipore, Bedford, MA, USA). The extracted luciferin was subjected to 1260 Infinity II (Agilent), using an Rx-C18 Analytical HPLC Column 4.6 ×250 (Agilent), eluted with a linear gradient of 10–70% MeOH in 80 mM Tris-HCl (pH 9.0) for 30 min at a flow rate of 1 mL/min. Fluorescence was detected (excitation at 383 nm; emission at 528 nm) with a 1260 Infinity fluorescence detector (Agilent). The retention time and peak area were analyzed using OpenLab CDS2 (Agilent). Under the foregoing conditions for HPLC analysis, luciferin was eluted at 21 min. Synthetic D-luciferin (Sigma–Aldrich) was used as the positive control.

### Protein extraction and western blot analysis
For protein extraction, whole bodies of male adults were sampled on the first day after emergence. The treatment groups were WT, *dsGfp*, *dsAlAbd-B*, *dsAlUnc-4*, and *dsAlPex13*. Specimens were frozen in liquid nitrogen, immediately ground, and stored at −80 °C. For a 100 mg sample, we rapidly added 300 μL of an ice-cold lysis buffer to the tube. The buffer comprised: 50 mM Tris (pH 7.4), 150 mM NaCl, 1% TritonX-100, 1% sodium deoxycholate, 0.1% SDS, 2 mM sodium pyrophosphate, 25 mM β-glycerophosphate, 1 mM EDTA, 1% Halt™ Protease Inhibitor Cocktail (Thermo Fisher Scientific Inc.), and 1 mM PMSF. This was followed by constant agitation for 2 h at 4 °C, and centrifugation for 20 min at 12,000 rpm at 4 °C in a microcentrifuge. After aspirating the supernatant, we placed it in a fresh tube, and kept it on ice. After discarding the pellet, we removed a small volume of the lysate to perform a protein quantification assay. After determining the protein concentration for each cell lysate, we determined the amount of protein to load, added an equal volume of the 2× loading sample buffer, and boiled the mixture at 100 °C for 5 min. After loading equal amounts of protein into the wells of the SDS-PAGE gel, we transferred the protein from the gel to the PVDF membrane. A blocking buffer was

used to block the membrane for 1 h at room temperature. The membrane was then incubated with 1000× dilutions of rabbit antibody to Firefly Luciferase (Bioss, bs-8539R), or mouse monoclonal antibody to α-tubulin (Beyotime, AT819) in a blocking buffer. The membrane was washed in three TBST washes, 5 min each, and then incubated with the 1000× dilution of goat anti-rabbit IgG-HRP (Beyotime, A0208) or goat anti-mouse IgG-HRP (Beyotime, A0216) in a blocking buffer at room temperature for 1 h. For signal development, we followed the BeyoECL Star kit (Beyotime, P0018) manufacturer's protocol. BIO-RAD Chemi-Doc™XRS+ was used for image acquisition and analysis.

### Immunofluorescence analysis (IF)
Immunofluorescence analysis was performed as previously described, with modifications[92]. Tissue sections were incubated with rabbit anti-Firefly Luciferase primary antibodies conjugated with Alexa Fluor 594 (Bioss, bs-8539R-AF594). For nuclear staining, the sections were stained with 4′,6′-diamidino-2-phenylindole (DAPI) and mounted with a fluorescence mounting medium (Agilent, CA, USA). Images were taken with an Eclipse C1 laser-scanning microscope (Nikon, Tokyo, Japan).

### Subcellular assays
To generate plasmids for subcellular assays, the CDS of *AlLuc1* and *AlLuc2* were separately cloned downstream from GFP of pcDNA3.1-EGFP at the *Xba*I site to construct the corresponding plasmids. Briefly, HEK293T cells seeded in 24-well plates were transfected with 300 ng of pcDNA3.1-EGFP-AlLuc1 or pcDNA3.1-EGFP-AlLuc2 plasmids. The GFP signal was excited at 488 nm and detected at 495–530 nm (color-coded as green). The nuclei were stained with DAPI (Thermo Fisher Scientific). The signals were visualized using the SP8 lightning confocal microscope system (Leica, Heidelberg, Germany)

### Yeast one-hybrid assays
Yeast one-hybrid assays were performed using the Matchmaker Gold Yeast one-hybrid Library Screening System (Clontech). The promoter sequence of *AlLuc1* (*proAlLuc1*) was amplified using the primers listed in Supplementary Data 3. Transcription start sites (TSS) were predicted using the Neural Network Promoter Prediction (this server runs the NNPP version 2.2 of the promoter predictor)[93]. The putative recognition motifs in the *AlLuc1* promoter (*proAlLuc1*) and two transcription factors (AlABD-B and AlUNC-4) were predicted using the Jaspar database (https://jaspar.genereg.net/)[94]. The criteria for identification of high-scoring motif regions were the existence of three or more interaction sites with an average interaction score higher than 0.85 within a sliding window of 100 bp. Two putative AlABD-B binding sites (−131 to −145 and −392 to −415 upstream of the TSS) were identified and deleted to generate the *proLuc1-mt1* mutant promoter. Two putative AlUNC-4 binding sites (−485 to −492 and −550 to −557) were identified and deleted to generate the *proLuc1-mt2* mutant promoter. *proAlLuc1*, *proLuc1-mt1*, and *proLuc1-mt2* were cloned into the Y1H vector pABAi using homologous recombination. CDS of *AlAbd-B and AlUnc-4* were separately cloned into the pGADT7 vector digested with *Nde*I and *Bam*HI. All plasmids were co-transformed into the Y1H Gold yeast strain following the manufacturer's instructions. The transformed cells were cultured in SD/-Leu/-Ura/AbA (with Aureobasidin A 0 ng/mL, 400 ng/mL, 500 ng/mL, 600 ng/mL) medium at 30 °C for 2 days. The empty pABAi vector and empty pGADT7 vector were used as the negative controls. The p53-ABAi vector and pGADT7-rec53 AD were used as positive controls.

### EMS assay
EMSA was carried out using the LightShift® Chemiluminescent EMSA kit (Thermo Fisher Scientific) according to the manufacturer's protocol. CDS of *AlAbd-B* was cloned into the pET28a vector digested with *Eco*RI and *Hind*III. The resultant plasmid was transferred into *E. coli*

BL21 (DE3), its expression was induced by 0.1 mM IPTG at 18 °C for 10 h, and the recombinant AlABD-B protein was purified using a Ni-NTA Sepharose Fast Flow column with elution buffer (NaH$_2$PO$_4$ 50 mM, NaCl 300 mM, Imidazole 300 mM; PH 7.2). The recombinant AlABD-B protein-saved buffer was changed to PBS with 10% glycerol using a 10 KDa Millipore Ultrafiltration centrifugal tube. Double-stranded oligonucleotide fragments (5′-ATA CAT TAA AGT ATC TAA TAA AAA TTA ATG GTT CAT TAA GAT ATA TAA AAA ACA-3′) containing ABD-B binding sites were biotinylated using an EMSA Probe Biotin Labeling Kit (Beyotime) and used as biotin-labeled wild-type probes (biotin-WT). The ABD-B binding site mutation probe was (5′-ATA CAT TAA AGT ATC cgg cgg gAA ccg gca aTc tgc cgg GAT ATA TAA AAA ACA-3′). The unlabeled WT probe and the ABD-B binding sit mutation probe were used as competitors to test binding specificity, and 2 μg/μL of Poly(dI-dC) was added as a nonspecific competitor. Protein (2.0 μg) and probes (0.2 pmol) were incubated for 60 min at room temperature, separated on a native 6% polyacrylamide gel at 100 V for 90 min, and then transferred onto a nylon membrane (Thermo Fisher Scientific) at 380 mA for 30 min. UV cross-link oligonucleotides were placed onto the nylon membrane, and images were acquired using darkroom development techniques for chem-luminescence using BIO-RAD Che-miDoc™XRS+.

### Dual-luciferase reporter assay
The dual-luciferase reporter assay is a very sensitive and convenient method for examining the transcriptional activity of a gene. To create Dual-Luc reporter constructs (Supplementary Fig. 12), the promoter sequences of *AlLuc1*, *proAlLuc1-mt1*, and *proAlLuc1-mt2* fragments were separately cloned into the pGL4.17 vector containing a luciferase reporter gene at the *Xho*I and *Bgl*II restriction sites. The transcription factors of *AlAbd-B* and *AlUnc-4* CDS were cloned separately into the pcDNA3.1 vector at the *Kpn*I and *Apa*I restriction sites driven by the CMV promoter. pRL-TK corresponding to the renilla luciferase reporter, and driven by the HSV-TK promoter, was used as a control for transfection efficiency. HEK293T cell line (Procell Life Science & Technology Co., Ltd, CL-0005) was authenticated by STR analysis. HEK293T cells were plated into 24-well plates at a density of 150,000 cells/well in Dulbecco's Modified Eagle's Medium (Corning) + 10% fetal bovine serum and grown overnight. The cells were transfected in triplicate with varying amounts of each plasmid (transfection mix included 600 ng total plasmid and 1.5 μL Lipofectamine 2000 in 0.1 mL OptiMEM) and grown 24 h at 37 °C with CO$_2$. At the end of incubation, the cells were harvested and lysed in 0.1 mL PLB buffer. Luciferase activity was measured using a commercial dual-luciferase assay system (Promega) on a EnVision multilabel plate reader (Perkin Elmer, Hopkinton, MA). The relative luciferase activity was calculated as the ratio of firefly luciferase to renal luciferase (fLUC/rLUC).

### BiFC
Bimolecular fluorescence complementation assays (BiFC) is a technology typically used to validate protein interactions in vivo. The constructs used for AlABD-B and AlUNC-4 interactions were as follows: the *AlAbd-B* CDS was cloned upstream of the pBiFC-VN173 vector and named AlABD-B-YFP$^N$, and the N-terminal fragment of the Venus fluorescent protein was fused to the C-terminus of AlABD-B. The *AlUnc-4* CDS was cloned upstream of the pBiFC-VC155 vector and named AlUNC-4-YFP$^C$, which means that we fused the C-terminal fragments of the Venus fluorescent protein to the C-terminus of AlUNC-4. We used the same method to construct the plasmids for protein interactions between AlPEX13 - AlPEX14, AlPEX5 - AlPEX14, AlPEX5 - AllLUC1, AlPEX14 - AlLUC1, and AlPXMP2 - AlLUC1. Combinations of recombinant plasmid and empty vector were used as negative controls. HEK293T cells were co-transferred with 400 ng of AlABD-B-YFP$^N$ and AlUNC-4-YFP$^C$; AlPEX13-YFP$^N$ and AlPEX14-YFP$^C$; AlPEX5-YFP$^N$ and AlPEX14-YFP$^C$; AlPEX5-YFP$^N$ and AlLUC1-YFP$^C$;

AlPEX14-YFP$^N$ and AlLUC1-YFP$^C$; AlPXMP2-YFP$^N$ and AlLUC1-YFP$^C$ (each) in a 24-well plate. The yellow signals of BiFC were visualized using the SP8 lightning confocal microscope system (Leica, Heidelberg, Germany). The nuclei were stained with DAPI (Thermo Fisher Scientific). DAPI-DNA fluorescence was measured using a 405 nm excitation filter and a 420–460 nm (blue) emission filter. Venus fluorescence was detected by recording emission at 525–570 nm (yellow) with the excitation wavelength set to 514 nm.

## Multicolor BIFC (mcBIFC)

By combining fragments of different fluorescent protein variants in mammalian cells, the original BiFC system was extended to multicolor BiFC (mcBiFC), which allows direct visualization of multiple protein interactions within a single cell[30]. Different mcBIFC reporter plasmids were constructed in vectors derived from the pCMV vector. First, *AlPex13* cDNAs was ligated upstream of a (GSSS)$_3$ linker and YN155 peptide (AlPEX13-(GSSS)$_3$-YN155) using the gene synthesis method, and subcloned into *Eco*RI/*Kpn*I restriction sites driven by the CMV promoter. AlPEX14-CC155 and AlPEX5-CN155 were constructed in the same manner. Full-length *AlLuc1* was amplified by PCR and subcloned into the pCDNA3.1-C-mCherry-Flag vector at the *Eco*RI/*Apa*I restriction sites. AlLUC1 was fused downstream from mCherry. All constructs were verified by sequencing. In the confocal imaging experiments, we used the same method as for BiFC, with minimal changes in plasmid concentrations. Cells were transfected in triplicate, with 800 ng of the total plasmid in each well.

## Statistical analyses

Statistical analyses were carried out using the Origin software (v8.5.0). Measurements were taken from distinct samples. Differences between samples were assessed using two-sided Student's *t*-tests, or one-way ANOVA. If significant differences were found, Tukey's multiple range tests were used to determine the differences between means. Data were expressed as mean ± standard error (SE), and $P < 0.05$ was the significance threshold.

## Reporting summary

Further information on research design is available in the Nature Portfolio Reporting Summary linked to this article.

## Data availability

Source data are provided with this paper. The genome assemblies and sequence data *A. leii* were deposited in NCBI under the BioProject accession number PRJNA948550, the whole genome project has been deposited at DDBJ/ENA/GenBank under the accession JARPUR010000000 [https://www.ncbi.nlm.nih.gov/nuccore/JARPUR000000000.1], and CNCB-NGDC (National Genomics Data Center, China National Center for Bioinformation) under the BioProject accession number PRJCA016073. Source data are provided with this paper.

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

## Acknowledgements

We thank Q. Liu, Z.P. Yao, and X.L. Li for breeding fireflies, and I. Jakovlić and L. Ballantyne for help in revising the grammar. National Natural Science Foundation grant of China (No. 32070485); National Key R & D Program of China (No. 2023YFD1400700).

## Author contributions

Conceptualization: X.F. and X.Z. Methodology: X.F. and X.Z. Visualization: X.F. and X.Z. Funding acquisition: X.F. Project administration: X.F. Supervision: X.F. Writing – original draft: X.F. and X.Z. Writing – review & editing: X.F. and X.Z.

## Competing interests

The authors declare no competing interests.
