## [Peer Review File · Nature Communications]

Key homeobox transcription factors regulate the development of the firefly's adult light organ and bioluminescenceREVIEWER COMMENTS

Reviewer #1 (Remarks to the Author):

Summary

The manuscript entitled "Key homeobox transcription factors regulate Firefly adult light organ development and bioluminescence" by Xinhua Fu and Xinlei Zhu aims at providing a better understanding of the molecular control of the light organ bioluminescence in adult fireflies. To this end, they used as a model *Aquatica leii*. They first employed genome sequencing to assemble more in-depth the genome. They next performed comparative analysis at the genome and transcriptome levels to characterize transcription factors (TFs) presence and expression level. Focusing on the homeodomain proteins, they used RNAi injection combined with expression and functional assays to assess their importance in pupae light organ formation and bioluminescence. They further focused on ALABD-B and ALUNC-4 and performed interaction and reporter assays in human cells and in vitro to propose a possible mechanism of ALUC1 regulation and peroxisome transport.

Strength

From genome assembly, comparative analysis, transcriptome and gene-function relationship, this manuscript is an integrative study on an interesting topic. The field is specific but the evolutionary aspect studied holds the potential for broader interest in the evolution and gene regulation communities.

Weakness

The problem of syntax makes the manuscript often difficult to read. Moreover, there is a lack of essential information and the description of the figures in text and legends is too superficial. This makes the reading often confusing (for example, male VS female, n number, data mentioned but not provided like Alap2-RNAi and videos). The result part is in particular lacking description and importantly, conclusion and reasoning. The discussion is missing.

Overall

While the story is interesting, with a lot of data, the dissonance of these data and of the methods employed as well as the lack of discussion, clarity and description impact strongly on the potential of the story.

In the reviewer's opinion, the present manuscript does not fulfil the requirement and accessibility necessary for publishing in Nature Communication. Thus, this reviewer cannot recommend it for publication.

Major issues

Results:

- Fig1D and 1A are not mentioned in the text
- page6 lines 141-142: how about the expression pattern in females?
- extended data fig5: how about the analysis for other TFs? What does the distribution look like?
- Interference efficiency: is it ubiquitous or tissue-specific? How was the qPCR performed? On whole individuals? Only the end of the abdomen?
- Fig2: ALAP2-RNAi: there is a decrease of intensity of bioluminescence (fig2A) in male but not in female. It would be interesting to understand why and provide quantification and reasoning to explain this discrepancy between males and females.
- page8: overall, what is the conclusion? What does it suggest? This seems missing
- Fig2B: on which tissue was the qPCR experiments performed?
- Fig2E is confusing and would gain visibility by changing the colour or pattern or size
- page12, line256: a sentence seems missing between "verified36" and "Analysis...", it seems to weaken the following description.
- page13, line 286: how about the protein level of luciferase?

- is it possible to perform RNAi of peroxisomal transporter and analyse the effect on the luciferase localisation?
- Fig4 and the following: Why using human cell lines as model? Why not insect cells? or staining of dissected abdominal segment?
- Fig4F: a negative control may strengthen the figure (mutant DNA sequence of the Abd-B binding site that should not compete)
- why doing YOH for UNC-4 and EMSA for ABD-B? this really confuses this reviewer. EMSA would have been sufficient for both proteins, + DNA mutant of each binding site for showing specificity.
- Is it possible to do a reporter assay in vivo within this specie? This would greatly enhance the quality of the study
- Fig5E and extended data fig12: it is rather confusing to detect BiFC signal of AIPEX14 and ALLuc1 in the nucleus. How would you explain this result?
- BiFC-related result: why not using tri-colour BiFC for showing the requirement of several subunits for ALLUC1 transport in the peroxisome? Related to this, this reviewer gets a bit lost as it seems that ALLUC1 expression was not revealed specifically in the peroxisome. May you clarify this please as it is major information for the manuscript?
- it would be interesting to test the requirement of the DNA binding ability of Unc-4 to interact and synergize with Abd-B activity on the reporter. One experiment could be to assess the transactivation of the reporter by the combinatorial expression of Unc-5 and Abd-B when a mutation of the Unc-4 binding site is inserted.

Reasoning missing: page8: it sounds like something is missing after "axongogenesis22"

Data/figure missing

- video and supplementary video missing
- extended data fig6: no flash behaviour present
- data AIShox2 knockdown is missing (page 8)
- quantification of fig5A is missing

Mix of figures:

- among many of them, fig1B cited before Fig1A, fig2A cited after 2B, fig 2C after 2D, fig4B before 4A, fig4E before 4D
- extended data fig7C not cited
- fig5E and extended data fig12 left panel are exactly the same

Legend lacking information:

- extended fig5, Fig2, Fig3A, Fig4 (and AF in particular), extended data fig6, extended data fig10, extended data fig11, extended data fig12

Discussion: missing

Minor issues

Abstract:

- present tense to uniformise
- page1, line 20: "our results demonstrate that ..."

Introduction:

- page2 line 39: "luciferaseS are" or "luciferase is"
- a short sum-up of the major finding would have been appreciated

Results:

- Page6, line 143: "only 6 homeobox genes were always.."
- Page7, line 150: quantitative PCR (not quantitation)
- Page7, line162 : corrective dash between Abd-B and specifies

Page10, ine 200-202: the syntax of the sentence is confusing, same line 215-216, 217-218, 227-228
Same page10,
Line 221: "were assessed" instead of "were detected" would provide better clarity
line 222, 226, 230: citing the fig 3B here would help the reader
page11, line 231: syntax is confusing
page12 line23: "functional" instead of "function"
page12: tense seems sometimes confusing (example line240: "Alluc1 is located" instead of "was")
page13, line 275: syntax "of Alluc1 was down-regulated.."

Reviewer #2 (Remarks to the Author):

This is an interesting study that provides new insight in the regulation of firefly luminescence. Given the availability of 4 already published firefly genomes, it is not clear why they needed to sequence a different one. Is it a better model system? Does it provide a different perspective on the problem? The genome sequencing, quality control and gene annotation follow standard protocols and provide the information required for the current study. The work is presented in a linear fashion and the overall meaning is clear. However the word usage and grammar need additional attention. The follow represent only a few instances, where the clarity of the argument was obscured.

Line 39 peroxisomes where luciferase..... peroxisomes containing luciferase.
Line 49 sifted should probably be filtered.
Line 165 no significant difference
line 215 and 217 is possibly

Line 224 sifted should be filtered
Line 267 in female abdomens among with female
Line 270 Alluc1 is needed for transport to must be transported to
Line 273 was interfered by was knock down by
Line 322 Need to included AIABDB in this sentence
Line 331 AIABD-B activated
Line 332 between in cells
Line 352 Alluc1 functions within in. and requires import by peroxins.
Line 354 sifted filtered
Line 355 were involved in import, RNAi was performed for each gene.
Line 361 Verification by real-time..... were down-regulated

Response to reviewer #1:

Summary

The manuscript entitled "Key homeobox transcription factors regulate Firefly adult light organ development and bioluminescence" by Xinhua Fu and Xinlei Zhu aims at providing a better understanding of the molecular control of the light organ bioluminescence in adult fireflies. To this end, they used as a model *Aquatica leii*. They first employed genome sequencing to assemble more in-depth the genome. They next performed comparative analysis at the genome and transcriptome levels to characterize transcription factors (TFs) presence and expression level. Focusing on the homeodomain proteins, they used RNAi injection combined with expression and functional assays to assess their importance in pupae light organ formation and bioluminescence. They further focused on *AlABD-B* and *AlUNC-4* and performed interaction and reporter assays in human cells and in vitro to propose a possible mechanism of *AILUC1* regulation and peroxisome transport.

Strength

From genome assembly, comparative analysis, transcriptome and gene-function relationship, this manuscript is an integrative study on an interesting topic. The field is specific but the evolutionary aspect studied holds the potential for broader interest in the evolution and gene regulation communities.

Weakness

The problem of syntax makes the manuscript often difficult to read.

Moreover, there is a lack of essential information and the description of the figures in text and legends is too superficial. This makes the reading often confusing (for example, male VS female, n number, data mentioned but not provided like *Alap2*-RNAi and videos)

The result part is in particular lacking description and importantly, conclusion and reasoning.

The discussion is missing.

Overall

While the story is interesting, with a lot of data, the dissonance of these data and of the methods employed as well as the lack of discussion, clarity and description impact strongly on the potential of the story.

Response: Thank you for your summary. We really appreciate your efforts in reviewing our manuscript. We apologize for the syntactic problems in the original manuscript and the problems they caused. The manuscript has been thoroughly revised, and parts of it rewritten where needed, by a native English speaker proficient in genetics, so we hope that the revised version can meet the journal's standard. We have also revised the manuscript according to other comments: (Male Vs Female), tests and analysis on female tissue were added; (n number), numbers of tested material were added in Methods part; (data mentioned but not provided like *Alap2*-RNAi), data mentioned like *Alap2* RNAi were added; Three Supplementary videos were uploaded; The result parts were re-written; The discussion part was added. Our point-by-point responses are detailed below.

Major issues

Results:

-Fig1D and 1A are not mentioned in the text

Response: Thank you for pointing out these problems: Fig1A is now mentioned in the Introduction, and Fig1D is mentioned in line 70.

-page6 lines 141-142: how about the expression pattern in females?

Response: During our sample collection at various developmental stages, our primary concern was securing a substantial number of individuals for dissection purposes. In addition, males also have a larger luminous organ. Therefore, we focused our sampling efforts and analyses solely on male specimens. To address your concerns, and analyze the expression patterns of these genes within the female luminous organ, we detected the relative expression level of 8 homeobox genes by qPCR in the light organ of female individuals at different developmental stages (n=3).

The relative expression levels of *AlAbd-A*, *AlAbd-B*, *AlUbx*, *AlAntp*, *AlUnc-4*, *AlShox2*, *AlRepo* and *AlAp2* were analysed in three developmental stages of female light organs using qPCR (Supplementary Fig. 5B). *AlAbd-B* and *AlUnc-4* were continuously up-regulated during the female pupal development, while *AlAbd-A*, *AlAntp* and *AlUbx* were down-regulated. The expression pattern of *AlAp2* had a peak expression in the mid-pupal stage, and the expression levels of *AlRepo* did not change during these three stages. Furthermore, the expression level of *AlShox2* could not be detected ($Cq > 37$) in female light organs. The results are shown in Supplementary Fig. 5b.

As fireflies undergo one reproductive cycle annually, we currently lack a sufficient female sample size for conducting more extensive RNA sequencing and analyses.

-extended data fig5: how about the analysis for other TFs? What does the distribution look like?

Response: In addition to the Homeobox transcription factor that we highlighted in yellow, the expression patterns of other transcription factors can be broadly categorized into two types: (1) down-regulated during the development of the light organ, and (2) peak expression in the mid-pupal stage. We hypothesise that these two types of transcription factors may be more important in the early stages of pupal development than in the late stages, or these genes may play an inhibitory role, but they don't affect the entire process of light organ development and they have no direct relationship with the control of bioluminescence and flashing. Therefore, they are not in the focus of this study. The results are shown in Supplementary Fig. 5A, and if the editor deems it necessary, we will add this explanation to the main manuscript.

-Interference efficiency: is it ubiquitous or tissue-specific? How was the qPCR performed? On whole individuals? Only the end of the abdomen?

Response: For each dsRNA treatment group, we used 2 males and 1 female. The total RNA was extracted from the whole individuals (ubiquitous). For the qPCR, we used three technical replicates. We have added the missing content to the line 783-984 of the Methods section.

-Fig2: *AlAP2*-RNAi: there is a decrease of intensity of bioluminescence (fig2A) in male but not in female. It would be interesting to understand why and provide quantification and reasoning to explain this discrepancy between males and females.

Response: Thanks for pointing out a phenomenon that we had overlooked originally. Based on your feedback, we have re-treated nine specimens (4 ♀ and 5 ♂) with ds*AIAP2*: 2 died, 6 exhibited a continuous glow phenotype, and one female exhibited a continuous glow phenotype and a significant decrease in light intensity. These results indicated that the reduction of intensity of bioluminescence is not a universal phenomenon. Additionally, we assessed the video data, and failed to observe a significant decrease in the brightness. This indicates that the observed decrease might be an artefact caused by random fluctuations in brightness. Based on the latest experimental results, we have replaced the male photo of *dsAIAP2* in Figure 2A.

-page8: overall, what is the conclusion? What does it suggest? This seems missing

Response: Thank you for underlining this deficiency. We have added the missing content to the manuscript on page 8: “Overall, based on the fact that the knockdown of homeobox genes *AIAbd-B* and *AIUnc-4* resulted in non-luminescence and empty peroxisomes, these two genes may be the key regulators required for normal light organ development. In addition, we found evidence that *AIAntp*, *AIRepo* and *AIAP2* are involved in flash control.”.

-Fig2B: on which tissue was the qPCR experiments performed?

Response: For each dsRNA treatment group, extracted total RNA from the whole body. We have added the missing content to the line 783-784 of the Methods section.

-Fig2E is confusing and would gain visibility by changing the colour or pattern or size

Response: Thank you for your suggestion. According to the reviewer’s comment, we have deleted Fig. 2D and Fig. 2E, and moved these data to Supplementary Fig. 6.

-page12, line256: a sentence seems missing between “verified36” and “Analysis...”, it seems to weaken the following description.

Response: Thank you for your comment. Due to the lack of depth of the PTS2-type protein in this research, we removed the discussion about this content completely.

-page13, line 286: how about the protein level of luciferase?

We deeply appreciate the suggestion. According to the reviewer's comment, we added a Western blot to detect the protein level of luciferase in different dsRNA-treated groups. The result was shown in Supplementary Fig. 8. The results indicate that treatment with *dsAIPEX13* did not affect the expression level of AILUC1, but the expression level of AILUC1 was significantly down regulated in the *dsAIAbd-B* and *dsAIUNC-4* treatment groups.

-is it possible to perform RNAi of peroxisomal transporter and analyse the effect on the luciferase localisation?

Response: Thank you for your suggestion. We followed it through and detected luciferase localisation in the adult light organ of *dsGfp*, *dsAlAbd-B* and *dsAlPex13* groups by the immunofluorescence (IF) method. Tissue sections were incubated with rabbit anti-firefly luciferase primary antibodies conjugated with Alexa Fluor 594.

The results showed that the signal density of LUC1 in the light organ of the *dsGfp* group was very high, filling each cell. Contrary to our expectation that there should be no signal of LUC1 in the *dsAlAbd-B* treated sample, the experimental results showed that after increasing the antibody incubation time, a discontinuous signal of LUC1 (red fluorescence) could be observed, with low signal brightness and many holes or gaps. In the *dsPex13* group, we could also detect the signals of luc1, but they were also discontinuous, which we suspect to be a reflection of small cavities caused by empty peroxisomes.

Supplementary Fig. 8C Immunofluorescence staining of localization anti-Luc1-AF594 antibodies in photogenic layer of *dsAlAbd-B* and *dsAlPex13*, Scale bar = 50 μ m, DAPI was used to stain nucleus.

-Fig4 and the following: Why using human cell lines as model? Why not insect cells? or staining of dissected abdominal segment?

Response: We did not use insect cells because the transformation system using sf9 cells in our lab exhibited some instability. Another reason for using human cells is because two related functional studies used mammalian cells (**reference 15**. Keller, G.-A., Gould, S., Deluca, M. & Subramani, S.

Firefly luciferase is targeted to peroxisomes in mammalian cells. Proceedings of the National Academy of Sciences 84, 3264-3268 (1987); **reference 25**. De Wet, J. R., Wood, K. V., DeLuca, M., Helinski, D. R. & Subramani, S. Firefly luciferase gene: structure and expression in mammalian cells. Molecular and cellular biology 7, 725-737 (1987)). When we followed the methods described in the reference literature, our results differed from theirs. In order to better understand this issue, we continued to use mammalian 293T cells for the detection system. The abdominal slices obtained after dissection were used for the IHC detection in this new experiment, aiming to display the tissue and cellular localization of the AILUC1 protein. However, regarding protein-protein interactions and protein-DNA interactions testing, we are unable to conduct these experiments in *A. leii* because there is no mature plasmid transient transfection technology for Coleoptera insects available.

-Fig4F: a negative control may strengthen the figure (mutant DNA sequence of the Abd-B binding site that should not compete)

Response: Based on your suggestion, we have added the mutated probe in the EMSA experiment and included the results in Supplementary Fig. 11B. The design of the mutated probe is shown in Supplementary Fig. 11C.

-why doing YOH for UNC-4 and EMSA for ABD-B? this really confuses this reviewer. EMSA would have been sufficient for both proteins, + DNA mutant of each binding site for showing specificity.

Response: The reason we did not use the EMSA method to test the interaction between AIUNC-4 and *proALLuc1* was because there were some problems in the process of expressing AIUNC-4 in prokaryotes. Therefore, we used the Y1H method to detect the interaction between protein and DNA. Based on your suggestion, we added an experiment using the Y1H method to test the interaction between AIABD-B and *pro-ALLuc1*, and used a mutated promoter as the negative control. The experimental results are shown in Fig. 4E. In addition, we included the EMSA result for AIABD-B in Supplementary Fig. 11A.

-Is it possible to do a reporter assay in vivo within this specie? This would greatly enhance the quality of the study

Response: Unfortunately, there is no mature transgenic technology for fireflies. So we couldn't do a reporter assay in vivo within this specie.

-Fig5E and extended data fig12: it is rather confusing to detect BiFC signal of AIPEX14 and AILuc1 in the nucleus. How would you explain this result?

Response: Thank you for your comment. Based on your suggestion, we conducted BiFC assays for AIPEX14 and AILUC1 again. The results show that the interaction is located in the nucleus and cytoplasm, The biological duplication results are shown in the following figure. We speculate that the problem may be due to our selection of an inappropriate image. Based on the latest experimental results, we replaced the BiFC images in Figure 5E.

Experimental data, not shown in this ms. Protein AIPEX14 and AILUC1 interaction analysis by BiFC in the 293T cells (Scale bar = 20 μ m).

-BiFC-related result: why not using tri-colour BiFC for showing the requirement of several subunits for AILUC1 transport in the peroxisome? Related to this, this reviewer gets a bit lost as it seems that AILUC1 expression was not revealed specifically in the peroxisome. May you clarify this please as it is major information for the manuscript?

Response: Thank you for underlining this deficiency. According to the information shown in the study suggested by the reviewer, a multicolor bimolecular fluorescence complementation (BiFC) assay was carried out. In this assay, the N-termini of the fluorescent proteins YFP (YN155) and CFP (CN155) were expressed with the C-terminus of CFP (CC155). When CC155 interacted with YN155, yellow fluorescence was produced, whereas when CC155 interacted with CN155, blue fluorescence was produced (Supplementary Fig. 14A). When pmCherry vector was fused with AILUC1 and expressed in 293T cells, the red fluorescence representing the AILUC1 protein could be found in both the nucleus and the cytoplasm. Following this, we fused AIPEX14 with CC155, AIPEX13 with YN155, and AIPEX5 with CN155 (Supplementary Fig. 14B), and co-expressed them together with mCherry-AILUC1 in 293T cells. YFP and CFP fluorescence emissions revealed that AIPEX14 interacted with both AIPEX13 and AIPEX5 (Supplementary Fig. 14C). More importantly, we found changes in the subcellular localization of AILUC1 protein in the co-expression group: most of the fluorescence was distributed in vesicular structures; Furthermore, in a few cells, we found that signals were distributed in a dotted pattern around the nucleus (Supplementary Fig. 14C, the bottom panel). We hypothesise that this might be the peroxisome.

Supplementary Fig. 14. AIPEX13, AIPEX14, AIPEX5, AIPXMP2 and AILUC1 interaction analysis by Multicolor BiFC in 293T cell. (A) Experimental basis of the multicolor assays. The AIPEX13:AIPEX14 heterodimer formation brings YN and CC together to form a yellow fluorescent protein (YFP) which could be detected at the YFP channel; the AIPEX5:AIPEX14 heterodimer formation brings CN and CC together to form an intact cyan fluorescent protein (CFP) protein which could be

detected using the CFP channel. (B) Vector constructs used in Multicolor BiFC. (C) Fluorescent images with the co-expression of indicated plasmids. The addition of AIPEX13, AIPEX14 and AIPEX5 changed the subcellular localization of AILUC1 (Scale bar, 20 μ m). The white arrow heads indicated colocalized signals. The white dashed area indicated the possible nuclei.

-it would be interesting to test the requirement of the DNA binding ability of Unc-4 to interact and synergize with Abd-B activity on the reporter. One experiment could be to assess the transactivation of the reporter by the combinatorial expression of Unc-5 and Abd-B when a mutation of the Unc-4 binding site is inserted.

Response: We repeated the Dual-Luc experiment using the plasmid construction method shown in Supplementary Figure 12. The new experimental results are shown in Figure 4G, while the old experimental results have been deleted. The new experimental results indicate that a combination of AIUNC-4 and AIABD-B produced higher expression levels than either transcription factor alone, which suggests that AIUNC-4 enhanced the activity of the *AILuc1* gene promoter together with AIABD-B.

Reasoning missing; page8: it sounds like something is missing after “axogenesis22”

Response: We appreciate the reviewer's suggestion. According to the reviewer's comment, we provided more details to describe the research background for the 8 homeobox genes. We added this to the Discussion section.

Data/figure missing

-video and supplementary video missing

Response: We apologize for the missing files in the original manuscript. We have re-uploaded the supplementary videos. Supplementary Video 1. Phenotype of *dsAlAbd-A*, *dsAlAbd-B*, *dsAlAntp*, *dsAlUnc-4*, *dsAlRepo*, *dsAlAp2*, *dsAlUbx* and *dsAlShox2*; Supplementary Video 2. Phenotype of *dsAILuc1*; Supplementary Video 3. Phenotype of *dsAlPex5*, *dsAlPex13*, *dsAlPex14* and *dsAlPxm2*.

-extended data fig6: no flash behaviour present

Response: Thank you for your suggestion. We have added the flash behaviour data in Supplementary Fig. 6.

-data AlShox2 knockdown is missing (page 8)

Response: We have added the *dsAlShox2* data in Supplementary Fig. 6 and Supplementary Video 1.

-quantification of fig5A is missing

Response: *AlPex13*, *AlPex14*, *AlPex5* and *AlPxmp2* resulted in a significant decrease in light intensity and continuous glow compared with the control. The flash behaviour data are shown in Supplementary Fig. 13. and Supplementary Video 3.

Mix of figures:

-among many of them, fig1B cited before Fig1A, fig2A cited after 2B, fig 2C after 2D, fig4B before 4A, fig4E before 4D

Response: We are very grateful to the reviewer for pointing out these problems. We have rearranged the citation order of these figures.

-extended data fig7C not cited

Response: The experimental data of extended data fig7C in the original manuscript showed that there was no significant change in the expression level of *AILuc2* after the treatment with *dsAILuc1*. However, as this conclusion has no direct relevance to this study, we have deleted this result.

-fig5E and extended data fig12 left panel are exactly the same

Response: Due to limitations in image size and space, negative controls are not included in Fig5E. Therefore, we showed the BiFC results and negative control results in the extended data fig12 in the original manuscript. For the revised manuscript, we have deleted the data underlying the extended data fig12 and showed it in the Source data.

Legend lacking information:

-extended fig5, Fig2, Fig3A, Fig4 (and AF in particular), extended data fig6, extended data fig10, extended data fig11, extended data fig12

Response: We appreciate the reviewer's suggestions. According to them, we added more details to these figure legends.

Discussion: missing

Response: We have added the Discussion section to the revised manuscript on page 26-32.

Minor issues

Response: We apologize for the mistakes in the manuscript. We made corrections according to the Reviewer's comments, and further had the manuscript proofread by competent reviewer.

Abstract:

-present tense to uniformise

Response: Some past tense were changed to present and uniformise.

-page1, line 20: "our results demonstrate that ..."

Response: changed to "our results show that....."

Introduction:

-page2 line 39: “luciferaseS are” or “luciferase is”

Response: The sentence changed to “The photocytes are abundant in mitochondria and peroxisomes containing luciferases.”.

-a short sum-up of the major finding would have been appreciated

Response: Thank you for the suggestion. We have added the short sum at the end of the Introduction (Lines 52, page 2). “In this study, we present a chromosome-level genome assembly for *Aquatica leii* Fu and Ballantyne, a rare aquatic firefly, using single-molecule Nanopore sequencing and high-throughput chromosome conformation capture sequencing technologies. We also present the functional studies and analyses of two key homeobox transcription factors regulating the adult light organ development in *A. leii*.”.

Results:

Page6, line 143: “only 6 homeobox genes were always..”

Response: Changed to “Only six homeobox genes were continuously up-regulated ($p < 0.05$) during the pupal development, while other genes were down-regulated or their regulation shifted from up-regulation to down-regulation during the pupal development (Supplementary Fig. 5A).”.

Page7, line 150: quantitative PCR (not quantitation)

Response: Changed to quantitative PCR.

Page7, line162 : corrective dash between Abd-B and specifies

Response: Incorrective dash has been deleted and the sentence was modified and moved to Discussion Section.

Page10, ine 200-202: the syntax of the sentence is confusing, same line 215-216, 217-218, 227-228

Response: The sentence (line 200-202) was changed to “To further study the regulation of the development of adult firefly light organs, we focused on AIABD-B and AIUNC-4 as putatively key transcription factors.”.

Sentences (line 215-218) were changed to “ Similarly, the expression of closely related genes, *Alpex3*, *Alpex16* and *Alpex19*, which might be related to the transportation of Peroxisome Membrane Protein (PMP)^{17, 18}, also decreased significantly in the *dsAlAbd-B* group. The expression of *ALP_{xmp2}*, possibly related to the nonspecific transportation of small molecules¹⁹, decreased significantly in *dsAlAbd-B* and *dsAlUnc-4* groups.”.

Same page10,

Line 221: “were assessed” instead of “were detected” would provide better clarity

Response: The word “detected” was changed to “assessed”.

line 222, 226, 230: citing the fig 3B here would help the reader

Response: Fig. 3B was cited.

page11, line 231: syntax is confusing

Response: The sentence (line 231) was changed to “Surprisingly, the expression level of *AlUnc-4* decreased significantly in response to the knockdown of *AlAbd-B* (Fig. 3B).”.

page12 line23: “functional” instead of “function”

Response: “function” was changed to “functional”.

page12: tense seems sometimes confusing (example line240: “*AlLuc1* is located” instead of “was”)

Response: past tense was changed to present tense.

page13, line 275: syntax “of *AlLuc1* was down-regulated..”

Response: Sentence was changed to “Verification of real-time qPCR confirmed that the expression level of *AlLuc1* was down-regulated significantly in non-luminescent adults (Supplementary Fig. 8A).”.

Response to reviewer #2:

This is an interesting study that provides new insight in the regulation of firefly luminescence. Given the availability of 4 already published firefly genomes, it is not clear why they needed to sequence a different one. Is it a better model system? Does it provide a different perspective on the problem?

Response: ①The genome of *A.leii* is the first sequenced aquatic firefly genome, while the other 4 already published firefly genomes are from terrestrial fireflies. ②We have been able to raise *A.leii* in large numbers in the lab, which will provide us with sufficient experimental materials such as pupae to study development of adult light organs. Thus make *A. leii* as a better model system to study adult light organ development and flash control. The other 4 species of fireflies are difficult to be raised, i.e they can only be collected in the wild.③ We used single-molecule Nanopore sequencing and Hi-C technologies for genome sequencing and assembly, with an N50 125Mb, resulting in a higher-quality genome compared to the other 4 genomes.

The genome sequencing, quality control and gene annotation follow standard protocols and provide the information required for the current study.

The work is presented in a linear fashion and the overall meaning is clear. However the word usage and grammar need additional attention. The follow represent only a few instances, where the clarity of the argument was obscured.

Response: Thank you for your careful review. We appreciate the reviewer's positive evaluation of our work. We are very sorry for the mistakes in this manuscript and for the inconvenience they caused. According to your advice, we amended the relevant parts of manuscript. In addition, the manuscript has been thoroughly revised and rewritten by a native English speaker, so we hope it can meet the journal's high standards.

-Line 39 peroxisomes where luciferase..... peroxisomes containing luciferase.

Response: Changed to “peroxisomes containing luciferases”.

-Line 49 sifted should probably be filtered.

Response: changed to “In this study, we present a chromosome-level genome assembly for *Aquatica leii* Fu and Ballantyne, a rare aquatic firefly, using single-molecule Nanopore sequencing and high-throughput chromosome conformation capture sequencing technologies. We also present the functional studies and analyses of two key homeobox transcription factors regulating the adult light organ development in *A. leii*.”.

-Line 165 no significant difference

Response: Changed to “ The depletion of *AIUbx* transcript resulted in pale yellow hindwings, but it did not produce any significant impacts on the development of adult light organs and flash behavior (Supplementary Fig. 6A-D and Supplementary Video 1).”.

-line 215 and 217 is possibly

Response: Sentences were rewritten to “ Similarly, the expression of closely related genes, *Alpex3*, *Alpex16* and *Alpex19*, which might be related to the transportation of Peroxisome Membrane Protein (PMP)^{17, 18}, also decreased significantly in the *dsAlAbd-B* group. The expression of *AlPxmp2*, possibly related to the nonspecific transportation of small molecules¹⁹, decreased significantly in *dsAlAbd-B* and *dsAIUnc-4* groups.”.

Line 224 sifted should be filtered

Response: “sifted” changed to “Candidate genes for more detailed analyses were selected using two criteria: high expression level (in the top 10% of genes) and the change in expression $-\log_2FC$ (*dsAlAbd-B/dsGFP*) > 1.5. The selected genes comprised *Alluc1*, *AlPx11c.1*, *AlPx11c.2*, *AlPx11c.2*, *AlPex5*, *AlPxmp2*, *AlPex13*, *AlPex14*, *AlPex16* and *AlPex1*.”.

Line 267 in female abdomens among with female

Response: The sentence was changed to “Whereas *AILuc1* was specifically expressed in light organs (Supplementary Fig. 7A), *AILuc2* had the highest expression in female abdominal tissue (excluding the light organ) (Supplementary Fig. 7B).”.

Line 270 *AILuc1* is needed for transport to must be transported to

Response: The sentence was changed to “To test our hypothesis that nonfunctional peroxisomes were directly caused by the lack of *AILUC1* protein, we applied RNAi to the *AILuc1* gene.”.

Line 273 was interfered by was knock down by

Response: “was interfered by” changed to “we applied RNAi to the *AILuc1* gene.”.

Line 322 Need to included *AIABDB* in this sentence

Response: The sentence was changed to “These results indicated that *AIABD-B* and *AIUNC-4* are likely to be upstream regulators of *AILuc1*.”.

Line 331 AIABD-B activated

Response: We re-wrote the sentence as “The AIUNC-4 alone activated the reporter gene expression above the basal levels (Fig. 4G), but AIAbd-B activated the reporter significantly higher than AIUNC-4.”.

Line 332 between in cells

Response: According to the comment of Reviewer1 “-it would be interesting to test the requirement of the DNA binding ability of Unc-4 to interact and synergize with Abd-B activity on the reporter. One experiment could be to assess the transactivation of the reporter by the combinatorial expression of Unc-4 and Abd-B when a mutation of the Unc-4 binding site is inserted.”.

We repeated the Dual-Luc experiment using the plasmids construction method shown in Supplementary Fig. 12. The new experimental results are shown in Figure 4G, while the old experimental results have been deleted. The new experimental results indicate that a combination of AIUNC-4 and AIAbd-B produced higher expression levels than either transcription factor alone, which suggests that AIUNC-4 enhanced the activity of the *AILuc1* gene promoter together with AIABD-B.

So we re-wrote the whole paragraph as below:

“Dual-luciferase reporter assays were carried out for further validation of AIABD-B and AIUNC-4 transcriptional activation of the *AILuc1* promoter. Different plasmids were constructed individually (Supplementary Fig. 12).

The AIUNC-4 alone activated the reporter gene expression above the basal levels (Fig. 4G), but AIAbd-B activated the reporter significantly higher than AIUNC-4. However, a combination of AIUNC-4 and AIAbd-B produced higher expression levels than either transcription factor alone, which suggests that AIUNC-4 enhanced the activity of the *AILuc1* gene promoter together with AIABD-B.

Cis-element mutants were also analyzed by using the *proAILuc1-mt1* (AIABD-B binding site mutant) and *proAILuc1-mt2* (AIUNC-4 binding site mutant) (Fig. 4G). Each mutant was compared to a positive control (the wild-type plasmid with the relevant combinations of effector plasmids). The separate mutation of each cis-element abolished the synergic activation (Fig. 4G, the reporters of *mt1* and *mt2* groups both significantly decreased compared with the wild-type group).”.

Line 352 *AILuc1* functions within in. and requires import by peroxins.

Response: The sentence was changed to “As we have demonstrated that luciferase *AILUC1* functions within the peroxisomes (Fig. 4D), this suggests that *AILUC1* requires certain peroxins for import into peroxisomes.”.

Line 354 sifted filtered

Response: “sifted” changed to “screened”.

Line 355 were involved in import, RNAi was performed for each gene.

Response: Changed to “We performed RNAi analyses to verify whether the screened peroxins (*AIPX11C.2*, *AIPXMP2*, *APEX5*, *APEX13*, *APEX14*, *APEX16* and *APEX1*) are involved in the import of *AILUC1*. ”.

Line 361 Verification by real-time..... were down-regulated

Response: Changed to “Verification by the qPCR confirmed that the expression levels of *AIPex13*,

AlPex14, *AlPex5* and *AlPmp2* were down-regulated significantly in *dsAlPex13*, *dsAlPex14*, *dsAlPex5* and *dsAlPmp2* adults respectively.”.

REVIEWERS' COMMENTS

Reviewer #1 (Remarks to the Author):

This reviewer thank the authors for addressing the issues step by step and improving the quality of the manuscript. The paper is in my opinion a new manuscript, so this reviewer has only evaluated the revised status.

Some extra comments are provided below:

Page 9:

Why talking only about the qPCR in the female? How about the male?

It is followed by the RNAi data, but it is not clear if it has been done in males or females.

Same Fig2B, it is still not clear or written if it is data from male or female (legend, text).

This needs further clarification in text and figure legend.

Fig4 YOH: mut1 & mut2 have different effects with alAbdB & AIUnc4 (fig4E-F) but this doesn't seem to correlate with the reporter assay (fig4G). Why?

There are still several syntax problems that should be addressed in particular in the introduction and discussion. Just a few examples are listed but more are included.

Line22 : syntax

Line46 developmen

Line 52-53 : fu ballantyne ?

Line 483: organs

Etc...

Line527: The posterior Hox genes do not only suppress the function or more anterior one, but they also repressed their transcriptional expression

Discussion: it would be interesting to discuss the decreased expression of unc4 with ds-abdB and how it would fit in the cartoon model.

Reviewer #2 (Remarks to the Author):

The arguments in the manuscript are much clearer. The use of this model system is improved with the new genome, as demonstrated by the analysis of the adult lantern organ.

Response to reviewer #1:

This reviewer thank the authors for addressing the issues step by step and improving the quality of the manuscript. The paper is in my opinion a new manuscript, so this reviewer has only evaluated the revised status.

Some extra comments are provided below:

Page 9:

Why talking only about the qPCR in the female? How about the male? It is followed by the RNAi data, but it is not clear if it has been done in males or females.

Answer: For this issue, we hope to explain it through four parts: experimental purpose, experimental design, experimental results, and supplementary experiments.

1. **Experimental purpose:** The experimental purpose was to identify key homeobox transcription factors among different light organ developmental stages by comparative transcriptome analysis, and apply them in subsequent functional validation studies.

2. **Experimental design:** It is very difficult to get pupae directly in the wild, and it is also difficult to raise fireflies in the laboratory. It is necessary to retain sufficient female adults for reproduction. Therefore, in our experimental design, we only selected the material of male pupae as the research object. The larger area of the male light organ is another key factor that we took into account.

3. **Experimental results:** From the final results of the experiments, we have screened several homeobox transcription factors related to the development of light organ, achieving our experimental purpose.

4. **Supplementary experiments:** Following the reviewer's comment, we made an effort to conduct additional experiments. We used qPCR (instead of the transcriptional method in the experimental design) to explore the expression patterns of 8 candidate homeobox transcription factors at different developmental stages in the female light organ (page 9 lines 155-161, and Supplementary Fig. 5B).

5. In the RNAi experiments, in order to ensure the reproducibility of the experiment, both sexes were analyzed, and the same or similar stable phenotypes were obtained in both males and females.

Finally, as suggested by the reviewer, we revised the text in this round of revision to specify the sex of specimens used for the experiments in the text and figure legends. In the discussion, we mentioned these problems briefly in the limitations paragraph: "Compared with other model insects, firefly breeding is relatively difficult, which limits our research. For example, transcriptomic data from different developmental stages are available for males, but not for females. In the follow-up study, we will strengthen the research on the differences between the two sexes in fireflies."

Same Fig2B, it is still not clear or written if it is data from male or female (legend, text).

This needs further clarification in text and figure legend.

Answer: We revised both the text and figure legend.

Fig4 Y1H: *mut1* & *mut2* have different effects with *AlAbdB* & *AlUnc4* (fig4E-F) but this doesn't seem to correlate with the reporter assay (fig4G). Why?

Answer: Thank you for your comment. *proAlLuc1* is a wild-type promoter, *proAlLuc1-mt1* is a mutant promoter with the deleted DNA binding motif *AlABD-B*, and *proAlLuc1-mt2* is a mutant promoted with the deleted DNA binding motif *AlUNC-4*. In the Y1H assay, there was no significant interaction between *mt1* and *AlABD-B* (Figure 4E), and no significant interaction between *mt2* and *AlUNC-4* (Figure 4F). In the Dual-Luc assay (Figure 4G), we also detected this phenomenon. In order to display this result more clearly, we have re-grouped and re-arranged the experimental data. The results are shown in the table below. According to the data in the table, the results of Y1H and Dual-Luc are consistent and there is no significant conflict between them.

Transcription factor	AlABD-B			AlUNC-4		
Promoter	proAlLuc1	proAlLuc1-mt1		proAlLuc1	proAlLuc1-mt2	
Motif description	Wild-type promoter	Deleted DNA binding motif of AlABD-B		Wild-type promoter	Deleted DNA binding motif of AlUNC-4	
Y1H	Interaction			No interaction		
Y1H figure	Figure 4E 			Figure 4F 		
Dual-Luc	High luciferase fold detection		Luciferase activity significantly reduced	High luciferase fold detection		Luciferase activity significantly reduced
Dual-Luc figure						

We added a brief discussion of this problem to the revised manuscript:

“However, according to the Dual-Luc results, mutations in promoters significantly reduced the interaction activity, but did not completely inhibit the expression of the reporter. The underlying reason for this may be that, even though high-scoring motif regions were deleted in *proAlLuc1-mt1* and *proAlLuc1-mt2* mutants, there might still exist low-scoring motif sequences within the promoter that have the potential to engage with *AlABD-B* or *AlUNC-4*.”.

There are still several syntax problems that should be addressed in particular in the

introduction and discussion. Just a few examples are listed but more are included.

Line22 : syntax

Line46 developmen

Line 483: organs

Etc...

Answer: Thank you for spotting these mistakes. Due to a huge amount of work in the last revision, we were in a hurry to submit the revised manuscript before the deadline, and apparently failed to proofread it with sufficient attention to detail. We apologise for this. During this revision, we proofread it very carefully, and fixed a number of formatting, spelling, and syntactic errors. We are confident that the manuscript is now written to a high standard.

Line 52-53 : fu ballantyne ?

Answer: Fu and Ballantyne are the authority for the binomial name of this firefly species (*A. leii*). We also added the year in the revision "*Aquatica leii* (Fu et Ballantyne 2006)" to make it clearer.

Line527: The posterior Hox genes do not only suppress the function or more anterior one, but they also repressed their transcriptional expression

Answer: We revised the text in the following way: "In several instances, multiple Hox genes are known to be expressed within the same segment⁴⁶; in such cases, posterior Hox genes suppress the function of comparatively anterior Hox genes, commonly by repressing their transcription. This phenomenon is known as posterior dominance⁴⁷."

Discussion: it would be interesting to discuss the decreased expression of unc4 with ds-abdB and how it would fit in the cartoon model.

Answer: We are grateful for the suggestion. We added a paragraph regarding this question to the Discussion section of the manuscript. We also added a Supplementary Fig. 16 and provided detailed explanations in the figure legend.

The paragraph in the Discussion is copy-pasted below: "In this study, we discovered the interaction between transcription factors AIABD-B and AIUNC-4 in the light organs of firefly *A. leii*. Interactions between transcription factors DmUNC-4, DmABD-A, and DmUBX were previously found in *Drosophila melanogaster* using the BiFC method⁵¹, but until now there has been no evidence of the interaction between UNC4 and ABD-B. Here we showed that after *dsAIAbd-B*, the adult light organs, commonly consisting of three layers (epidermis (EP), photogenic layer (PL), and dorsal layer (DL)), exhibited irregular morphology and were not luminescent. After *dsAIUnc-4*, the structure of the PL layer was significantly changed in adult light organs (accompanied by non-luminescence), while EP and DL layers did not exhibit major changes. Hence, we proposed a theory about the mechanism of the development of the light organs (Supplementary Fig. 16). AIABD-B regulates the development of the entire adult light organ, and its RNAi results in a darkened cuticle, disruption of the bioluminescent layer's peroxisomes, and loss of the DL layer. AIUNC-4 participates in the development of the light organs and regulates the development of peroxisomes in photocytes by interacting with AIABD-B. Reduced expression of AIABD-B or AIUNC-4 leads to the malformation of

peroxisomes, a reduced number of peroxisomes, mislocation of the matrix proteins of peroxisome (such as AILUC1 etc.), and interruption of the normal light emission by light organs.”.

Supplementary Fig. 16. AIABD-B and AIUNC-4 are required for the development of the adult light organ

and peroxisomes within it. During the development of the adult light organ in the pupal stage, the key

transcription factor AIABD-B regulates the development of the light organ. AIABD-B may suppress the transcriptional expression and function of AIABD-A, suppress the synthesis of melanin, and cause transparency of the epidermis (EP) of light organs. AIABD-B also regulates the synthesis and transport of uric acid, which partakes in the development of the dorsal layer (DL). The development of peroxisomes in photocyte is also regulated by AIABD-B which partakes in the development of the photogenic layer (PL). A key transcription factor, AIUNC-4, participates in the development of PL. AIUNC-4 regulates the development of peroxisomes by interaction with AIABD-B. Decreased or absent expression of AIABD-B possibly results in the expression of AIABD-A, promotes the synthesis of melanin and darkens EP. A decrease in AIABD-B levels (or its absence) suppressed the synthesis and transport of uric acid and resulted in the destruction or absence of the entire dorsal layer. The absence of AIABD-B or AIUNC-4 resulted in deformed peroxisomes and non-luminescence. In the schematics, active genes are indicated in green coloring, inactive genes are indicated in gray coloring. bold lines indicate functional pathways, and thin lines indicate non-functional pathways. Lines terminating with an arrowhead indicate regulation in which the transcription factor functions as an activator, and lines terminating in a nail-head shape indicate repression. *AIAbd-B* or *AIUnc-4* RNAi lead to the pathway loss of function is indicated in red X.

Response to reviewer #2:

The arguments in the manuscript are much clearer. The use of this model system is improved with the new genome, as demonstrated by the analysis of the adult lantern organ.

Answer: We appreciate the reviewer's positive evaluation of our work.